# The ACUTE Protocol: Operationalizing Language Model Activations for Better Calibration, Utility, and Trust

**Nishant Subramani** [1] [2]  **Palash Goyal** [2]  **Yiwen Song** [2]  **Mani Malek** [2]  **Yuan Xue** [3]  **Tomas Pfister** [2]  **Hamid Palangi** [2]

## Abstract

As language models improve and become increasingly deployed to solve a variety of tasks, trustworthiness becomes essential. Calibration is a good proxy for trust: well-calibrated confidence estimates help inform the risk versus reward trade-off when trusting a specific model output. Unfortunately, even as models improve, they remain poorly calibrated, often biasing towards overconfidence. Additionally, calibration can be gamed: a policy that always predicts the base rate is perfectly calibrated, but completely uninformative. To resolve this, we develop a new metric, expected utility renormalized by the oracle (EURO), that balances calibration and informativeness. We also propose a general-purpose **a**ctivation-based **c**onfidence, **u**tility, and **t**rust **e**stimation protocol (ACUTE) to appropriately adjudicate uncertainty. The ACUTE protocol provides flexible, sample-efficient, and compute-efficient confidence estimators for 3 tasks including multiple choice question answering, tool-calling, and scientific document summarization across 6 models from 4 model families. ACUTE outperforms strong baselines on EURO, while maintaining low calibration error. Taken together, our work shows that equipping LLMs with the ACUTE protocol can improve calibration, utility, and trustworthiness in numerous settings.

## 1. Introduction

Users increasingly rely on large language models (LLMs) for many tasks such as information seeking, writing, and agentic tool-calling. Many technological solutions integrate LLMs into their workflow at many stages, trusting and using model output for further computation. As a result, automatically knowing whether to trust a model output is essential.

A confidence estimator can be an auxiliary model that estimates the probability that the output of another model (*e.g.*, an LLM) is correct. *Well-calibrated* confidence estimates can provide a signal of whether to trust a generation and what the associated risks could be. Practitioners often rely on a simple confidence estimator to evaluate whether to trust an LLM output: the probability the LLM itself assigns to the generation (*i.e.*, the product of token probabilities), even though this is known to be *poorly calibrated* in both single- and multi-token generation settings (Desai & Durrett, 2020; Jiang et al., 2021; Zhong et al., 2023; Stengel-Eskin & Van Durme, 2023a; Subramani et al., 2025a).

Ideally, we want a metric to adjudicate how much to trust a model generation, while accounting for how risky the task is (*i.e.*, for a high risk task we require more certainty to trust an LLM output than a low risk task). Calibration, often measured using expected calibration error (ECE; Murphy & Epstein, 1967; Naeini et al., 2015), solely measures how well predicted probabilities match observed outcomes. This has *two major drawbacks* for trust:

*ECE cannot distinguish between an oracle and a perfectly uninformative estimator that just predicts the base rate for all outputs independent of correctness.* Suppose we have a task that an LLM gets exactly 50% accuracy on. Both an oracle estimator (assigns a probability of 1 to correct and 0 to incorrect outputs) and a baserate estimator (assigns the base rate of 0.5 to all outputs) achieve an ECE of 0 (Figure 1).

*ECE is invariant to the associated risk of a task.* Suppose there exist two tasks with different risks levels: high risk requiring 0.9 probability to trust and medium risk requiring only 0.5 to trust and a probabilistic oracle (*e.g.*, one that assigns 0.75 to correct and 0.25 to incorrect outputs). This estimator gets an ECE of 0.25 for both tasks, even though it perfectly solves the medium-risk task, and fails on the high risk task (never trusting a single output).

Our contributions in this paper are:

1. A new general and easily interpretable metric called **expected utility renormalized by the oracle** (EURO), which balances calibration and decision-making util-

[1]Carnegie Mellon University [2]Google [3]Scale AI. Correspondence to: Nishant Subramani <nishant2@cs.cmu.edu>.

*Proceedings of the 43rd International Conference on Machine Learning*, Seoul, South Korea. PMLR 306, 2026. Copyright 2026 by the author(s).

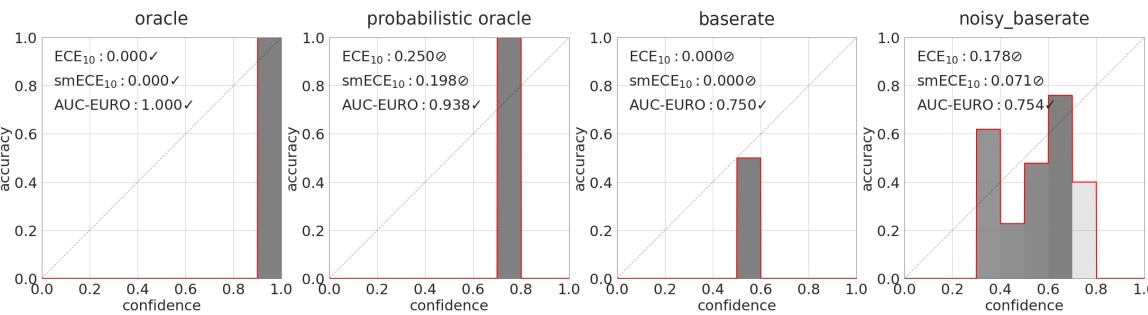

*Figure 1.* Here we show four different confidence estimators in order of decreasing quality and how the metrics ECE, smECE, and our metric AUC-EURO adjudicate them. Both ECE and smECE are errors so they should start at 0 and increase, while AUC-EURO is a utility so it should decrease. ECE and smECE break and decide that the two worst systems (baserate and noisy_baserate) are *much* better than the probabilistic oracle. AUC-EURO appropriately ranks them. Details of what each estimator are in §2.2.

ity. EURO resolves both drawbacks of ECE, granting practitioners a more reliable trust estimator (§2).

2. An **activation-based confidence, utility, and trust estimation protocol** (ACUTE) to assess the confidence of language model outputs (§3). Experiments on multiple-choice question answering (MMLU; Hendrycks et al., 2021), tool-calling (APIGen; Liu et al., 2024), and scientific document summarization (SCITLDR; Cachola et al., 2020) show that applying the ACUTE protocol on 6 LLMs improves confidence estimation via EURO, while maintaining low calibration error.

## 2. Expected Utility Renormalized by the Oracle (EURO)

### 2.1. Calibration, ECE, and Smooth ECE

Calibration measures how well predicted probabilities match the observed outcomes for a task. To measure how good a confidence estimator is for a task, expected calibration error (ECE) is often used (Murphy & Epstein, 1967; Dawid, 1982; Naeini et al., 2015; Naeini & Cooper, 2016). To calculate ECE, we build a histogram where bins correspond to predicted confidence bands (*e.g.*, 3 bins equate to 0.0 - $\frac{1}{3}$, $\frac{1}{3}$ - $\frac{2}{3}$, $\frac{2}{3}$ - 1.0). For every example $i$ within a bin, the accuracy of that example is compared to the estimator's predicted confidence: $|acc_i - \hat{p}_i|$. All of these absolute differences are averaged to produce a final ECE score. The number of bins can be challenging to set and different choices lead to different calibration scores.

Błasiok & Nakkiran (2024) introduce smooth ECE (smECE) a hyperparameter-free method to remove the reliance on the number of bins. [1] SmECE smooths out the histogram using Nadaraya-Watson kernel regression and measures calibration more precisely than ECE, but suffers from the same major drawbacks that ECE does. We introduce a metric that balances calibration with informativeness called ex-

pected utility renormalized by the oracle (EURO) to assess the decision-making utility of a confidence estimator. EURO has *just one* parameter: the normalized net utility of correctly abstaining from trusting an incorrect generation ($u_{ca}$).

### 2.2. Calibration Alone is Insufficient

Suppose for some task, we have language model $M$ that outputs predictions $y_i, \ldots, y_n$ and has 50% accuracy on these (*i.e.*, $\sum_i^n y_i = \frac{n}{2}$). Consider the following estimators:

1. oracle (1 if $y_i$ is correct and 0 otherwise)
2. probabilistic oracle (0.75 correct, 0.25 incorrect)
3. baserate (0.5 regardless of what $y_i$ is)
4. noisy baserate ($p \sim U(0.25, 0.75)$ regardless of $y_i$)

The oracle is perfect and the probabilistic oracle is indistinguishable from the oracle as long as the probability threshold $\tau \in (0.25, 0.75]$ (*i.e.*, for $\hat{p}$ emit by the estimator, trust if $\hat{p} > \tau$). Both base rate and noisy base rate estimators are uninformative for decision-making: they generate middling probabilities that are uncorrelated with correctness.

In Figure 1, we plot the reliability diagrams for these four estimators and their associated ECE, smECE, and our metric AUC-EURO, and find two concerning trends with both ECE and smECE. [2] First, they cannot distinguish between the oracle estimator and the perfectly uninformative base rate estimator (ECE=smECE=0). Our metric, AUC-EURO, however, does distinguish between them (oracle gets a score of 1, while the baserate estimator gets a score of 0.75). Second, the probabilistic oracle gets the worst performance score on ECE and smECE despite being a great confidence estimator. AUC-EURO appropriately scores this as worse than the true oracle, but much better than either the baserate or noisy baserate estimators.

Both ECE metrics have a third drawback: they are invariant to the associated risk of a task. Consider two tasks with dif-

---

[1] Kernel width is determined automatically from data.

[2] See §2.3 for details on how to calculate AUC-EURO.

ferent risk levels: a high risk task (*e.g.*, evaluating a trading agent that can buy and sell equities in a stock market) and a medium risk task (*e.g.*, scientific document summarization to use in a literature review for a course project). For the high risk task, we must be very sure the output is correct before trusting the agent to execute a trade; lets suppose that the threshold $\tau$ to trust a generation is 0.9 (*i.e.*, $\hat{p} > 0.9$ to trust). For the medium risk task, lets assume that $\tau = 0.5$. The probabilistic oracle estimator is good at abstaining for the high risk task, but it never trusts the output because all probabilities are less than 0.9. An ideal metric should give the estimator a lot of credit for abstaining often, but with some penalty for never trusting. For the medium risk task however, the probabilistic oracle estimator is perfect. Since both calibration error metrics are invariant to risk level, the estimator gets the same scores regardless of task. EURO, on the other hand, incorporates risk level; the probabilistic oracle achieves EURO = 0.9 and EURO = 1.0 for the high and medium risk tasks respectively.

### 2.3. EURO

For a given query $q$, a language model $M$ generates a candidate solution $\hat{y}$; a confidence estimator $C$ assigns a probability $\hat{p}$ to the example. There are four outcomes based on the Minimum Bayes Risk (MBR) decision:

1. True Positives (tp) := $\hat{p} > \tau$ and $\hat{y}$ is correct
2. False Positives (fp) := $\hat{p} > \tau$ and $\hat{y}$ is incorrect
3. False Negatives (fn) := $\hat{p} \leq \tau$ and $\hat{y}$ is correct
4. True Negatives (tn) := $\hat{p} \leq \tau$ and $\hat{y}$ is incorrect

Each of these four outcomes has an associated reward or utility: $R_{tp}$, $R_{fp}$, $R_{fn}$, $R_{tn}$ that can be set by the user based on the task at hand. Given the number of true positives, false positives, false negatives, and true negatives $(N_{tp}, N_{fp}, N_{fn}, N_{tn})$, the total utility $\mathbb{U} = R_{tp} * N_{tp} + R_{fp} * N_{fp} + R_{fn} * N_{fn} + R_{tn} * N_{tn}$. As a result, the MBR decision threshold for $\hat{p}$ is defined as: [3]

$$\hat{p} > \tau \stackrel{\text{def}}{=} \frac{R_{tn} - R_{fp}}{(R_{tp} - R_{fn}) + (R_{tn} - R_{fp})} \quad (1)$$

In our setting, the confidence estimator trusts the output generation if $\hat{p} > \tau$, and abstains if $\hat{p} \leq \tau$ because it is Bayes-optimal to do so. Since a confidence estimator should get a greater reward for trusting when the generation is correct and abstaining when incorrect than the opposite, $R_{tp} > R_{fn}$ and $R_{tn} > R_{fp}$. [4] Setting four values that have large ranges can be challenging, unintuitive, and difficult

---

[3]See the Appendix §A for a detailed derivation.

[4]Often $R_{tp}, R_{tn} \in [0, \infty)$ and $R_{fn}, R_{fp} \in (-\infty, 0]$ because getting a classification decision correct generally yields a positive reward and getting it incorrect yields at most 0 reward. In a standard binary classification task $R_{tp} = R_{tn} = 1$ and $R_{fp} = R_{fn} = 0$ and thus $\tau = 0.5$.

to analyze. To remedy this, we introduce two intuitive and tightly bounded quantities that help reparametrize this setup without loss of generality to have *just one* degree of freedom.

**Reparametrization:** We are interested in comparing confidence estimators $C_1$ and $C_2$ on the outputs of the same language model $M$ on the same dataset $D$. Crucially, $M$ generates $\hat{y}_1 \ldots \hat{y}_n$ and $C_1$ and $C_2$ *both* estimate the confidences of these candidate generations. $R_{tp}$, $R_{tn}$, $R_{fp}$, and $R_{fn}$ are based on the dataset, not the LLM or confidence estimator. When comparing $C_1$ and $C_2$ and the probabilities they emit, we only care about $\tau$ from equation 1.

**Net Utility of Correctly Trusting ($U_{ct}$):** When the model generates a correct answer ($\hat{y}$ is correct), we want our confidence estimator to emit $\hat{p} > \tau$. We assign a reward $R_{tp}$ or $R_{fn}$ based on whether $\hat{p} > \tau$ or not. Recall that $R_{tp} > R_{fn}$. Our reformulation looks at their difference, the net utility of trusting a correct generation.

**Net Utility of Correctly Abstaining ($U_{ca}$):** Similarly when $M$ generates an incorrect answer, we want the confidence estimator $C$ to output $\hat{p} \leq \tau$ and assign $R_{tn}$ or $R_{fp}$ accordingly. We look again to their difference, the net utility of abstaining from trusting an incorrect generation.

$$U_{ct} \stackrel{\text{def}}{=} R_{tp} - R_{fn}; U_{ca} \stackrel{\text{def}}{=} R_{tn} - R_{fp}$$

**Normalizing and Recomputing the MBR Threshold:** To simplify further, we calculate a normalized version of both utilities $u_{ct}$, $u_{ca}$ that are bounded between 0 and 1:

$$\begin{bmatrix} u_{ct} \\ u_{ca} \end{bmatrix} = \begin{bmatrix} U_{ct} \\ U_{ca} \end{bmatrix} \cdot \frac{1}{U_{ct} + U_{ca}} \quad (2)$$

Using $u_{ct}$ and $u_{ca}$, equation 1 can then be rewritten as: [5]

$$\hat{p} > \tau \stackrel{\text{def}}{=} \frac{U_{ca}}{U_{ct} + U_{ca}} = u_{ca} \quad (3)$$

We can now use $\tau$ as precisely the risk level associated with a task (*i.e.*, high risk means that an estimator gets a larger penalty for a false positive than reward for a true positive).

**Calculating the Relative Utility:** To calculate the relative utility between two confidence estimators $C_1$ and $C_2$, remember that the rewards ($R_{tp}$, $R_{tn}$, $R_{fp}$, and $R_{fn}$) and language model generations ($\hat{y}_1, \ldots \hat{y}_n$) are fixed. The only differences are the confusion matrices. The relative utility between two systems is the difference in total utility, $\mathbb{RU}_{C_1,C_2} = \mathbb{U}_{C_1} - \mathbb{U}_{C_2}$. The fixed generations ensure that the number of correct and incorrect samples are preserved: $N_{tp,C_1} + N_{fn,C_1} = N_{tp,C_2} + N_{fn,C_2}$ and

---

[5]Given the normalization scheme we chose, $u_{ct}$ = 1 - $u_{ca}$.

$N_{tn,C_1} + N_{fp,C_1} = N_{tn,C_2} + N_{fp,C_2}$. Thus, simplifying and normalizing with respect to $u_{ca}$ and $u_{ct}$ yields: [6]

$$\mathbb{RU}_{C_1,C_2} = \frac{1}{U_{ct} + U_{ca}} \cdot \begin{bmatrix} u_{ct} \\ u_{ca} \end{bmatrix} \cdot \begin{bmatrix} (N_{tp,C_1} - N_{tp,C_2}) \\ (N_{tn,C_1} - N_{tn,C_2}) \end{bmatrix} \tag{4}$$

**Renormalizing with respect to the Oracle and Anti-Oracle:** We want to put relative utilities on a scale with respect to the Oracle policy ($O$), a system that only gets true positives and true negatives, and the Anti-Oracle policy ($AO$), a system that only gets false positives and false negatives. Since the Oracle policy should get the highest possible utility and the Anti-Oracle policy get the lowest possible utility, we normalize the utility of an estimator accordingly: $\text{EURO}_C(u_{ca}) \in [0,1]$, approaching 0 as $C$ gets closer to the $AO$ policy and approaching 1 as $C$ gets closer to the $O$ policy. For a confidence estimator $C$, EURO is:

$$\text{EURO}_C(u_{ca}) = \frac{N_{tp,C} + u_{ca} \cdot (N_{tn,C} - N_{tp,C})}{N_{tp,O} + u_{ca} \cdot (N_{tn,O} - N_{tp,O})} \tag{5}$$

**Area under the EURO curve** Since EURO is parametrized by $u_{ca}$, each value is a setting for the net utility of correctly abstaining from trusting an incorrect candidate. We can compute the EURO for each of these, which traces a curve. To compare confidence estimators across all possible settings, we summarize EURO by taking the area under this curve and term this AUC-EURO. Since EURO $(u_{ca}) \in [0,1]$, AUC-EURO $\in [0,1]$. [7]

**Analyzing EURO across risk levels:** In equation 3, we observe that $\tau \stackrel{\text{def}}{=} u_{ca}$, so we split $u_{ca}$ into three risk bins, where low corresponds to $u_{ca} \in (0, \frac{1}{3})$, medium to $u_{ca} \in [\frac{1}{3}, \frac{2}{3})$, and high to $u_{ca} \in [\frac{2}{3}, 1)$. For the highest risk tasks, we should be most weary of trusting a generation and get the most credit for correctly abstaining from trusting an incorrect generation. For the lowest risk tasks, we should be the least weary of trusting a generation, but as a result, we get the least credit for correctly abstaining from trusting an incorrect generation. The threshold $\tau$ and thus $u_{ca}$ exactly determines how much confidence we require to trust a generation. We measure a total of 4 AUC-EURO values: one corresponding to the entire spectra of $u_{ca}$ values and one for each of the 3 risk-split settings mentioned above. [8]

## 3. ACUTE

Activation-based confidence estimation leverages the activations of the language model while generating a candidate

---

[6]See Appendix §A.3 for all the EURO derivations.

[7]AUC metrics are used widely in science (Wagner & Ayres, 1977; Geifman et al., 2019; Subramani et al., 2025b, *inter alia*).

[8]See §B for a discussion on how EURO relates to other calibration and decision-making utility tools.

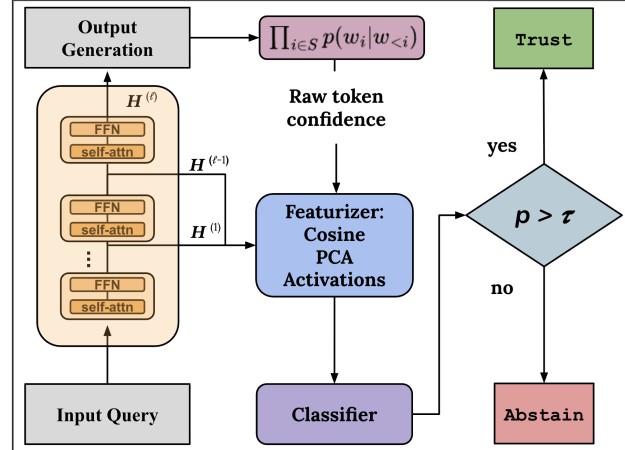

*Figure 2.* The ACUTE Protocol. We use input features from model internals to train confidence estimators to adjudicate whether to trust or abstain from trusting a model output. Details are in §3.

answer to learn a confidence estimator. In particular, the activations are used as input features to a classifier that predicts whether the resulting generation is correct or not. The probability that the classifier assigns to the "correct" class *is* the new calibrated confidence. Given that the latent spaces of language models are steerable via steering vectors (Subramani et al., 2022), there exist circuits and individual directions in latent space that are controllable and task specific (Li et al., 2023; Conmy et al., 2023; Syed et al., 2024; Li & Subramani, 2026b), and that activations at different layers of models correspond to different linguistic phenomenon (Tenney et al., 2019; Ethayarajh, 2019; Li & Subramani, 2026a), we hypothesize that the activations contain decipherable signals for confidence estimation.

### 3.1. Input features

**Mean-pooled Activations** Since we suspect that the activation spaces contain decipherable information for confidence estimation, the raw activations themselves are the most natural place to start. Consider a language model $M$ that processes an input query $q$ and produces a candidate generation $\hat{y} = w_1, \ldots, w_T$. At each layer $j$, we extract the activations for the output sequence: $H^{(j)} = \begin{bmatrix} \mathbf{h}_1^{(j)} & \ldots & \mathbf{h}_T^{(j)} \end{bmatrix}$. We collapse this activation into a single vector by mean pooling across the sequence such that for every layer $j$, we have a single activation vector per query: $\bar{\mathbf{h}}^{(j)}$. These become the input features to our confidence estimators. We train separate confidence estimators at each layer, but present the performance of the best performing estimator at different layer depths (early, middle, and late) rather than at all layers for simplicity.

**Layer-wise Cosine Similarities with the Final Layer** Activations at a single layer are high-dimensional and

| | MMLU | | | | | APIGen | | | | | SCITLDR | | | |
| | **smECE** (↓) | **AUC-EURO** (↑) | | | | **smECE** (↓) | **AUC-EURO** (↑) | | | | **smECE** (↓) | **AUC-EURO** (↑) | | | |
| estimator | | low | med | high | all | | low | med | high | all | | low | med | high | all |
|---|---|---|---|---|---|---|---|---|---|---|---|---|---|---|---|
| Raw Conf | 0.17 | 0.90 | 0.71 | 0.54 | 0.72 | 0.22 | 0.28 | 0.53 | 0.80 | 0.53 | 0.15 | 0.36 | 0.71 | **0.92** | 0.66 |
| HRE | 0.11 | 0.87 | 0.72 | 0.71 | 0.77 | **0.02** | 0.82 | 0.70 | 0.82 | 0.78 | **0.08** | 0.67 | 0.71 | **0.92** | 0.77 |
| NWKR | **0.07** | 0.90 | 0.73 | 0.74 | 0.79 | **0.02** | 0.82 | 0.69 | 0.82 | 0.78 | **0.08** | 0.67 | 0.71 | **0.92** | 0.77 |
| ACUTE early act | **0.07** | **0.91** | 0.73 | 0.74 | 0.79 | 0.05 | 0.90 | 0.82 | 0.87 | 0.86 | **0.08** | 0.69 | 0.72 | **0.92** | **0.78** |
| ACUTE mid act | **0.07** | **0.91** | 0.76 | 0.78 | 0.82 | 0.06 | 0.90 | **0.84** | **0.88** | 0.87 | **0.08** | **0.70** | 0.72 | **0.92** | 0.78 |
| ACUTE late act | **0.07** | **0.91** | **0.77** | **0.80** | **0.83** | 0.07 | 0.90 | **0.84** | **0.88** | 0.87 | **0.08** | 0.69 | 0.72 | **0.92** | 0.77 |
| ACUTE cosine | 0.09 | 0.90 | 0.75 | 0.78 | 0.81 | 0.03 | 0.87 | 0.77 | 0.84 | 0.83 | **0.08** | 0.68 | 0.71 | **0.92** | 0.77 |
| ACUTE pca10 | 0.08 | **0.91** | 0.76 | 0.78 | 0.82 | 0.04 | 0.90 | 0.82 | 0.87 | 0.87 | 0.09 | **0.70** | 0.72 | **0.92** | 0.78 |
| ACUTE pca20 | 0.08 | **0.91** | 0.76 | 0.77 | 0.81 | 0.06 | **0.91** | **0.84** | **0.88** | **0.88** | 0.09 | **0.70** | 0.72 | **0.92** | 0.78 |

*Table 1.* Results on the MMLU test set averaged across all 57 subtasks (left), on the APIGen test subset (middle), and on the SCITLDR dev set (right). All results for all tasks are averaged across the 6 LLMs we test. Lower smECE is better, while higher AUC-EURO is better. **Bold** indicates the best result and underline indicates the second best result in each column.

can be limited in the information it could contain. Layers tend to specialize; activations at earlier layers are better at probing word or subword-level phenomenon and those at layer layers are better at higher-level, sequence-level phenomenon (Rogers et al., 2020). Since concatenating the representations from all layers is infeasible as the number of input features would be too large (*e.g.*, 200,000 input features for a model with a hidden size of 4000 and 50 layers), we choose an aggressive reduction. For a model with $\ell$ layers, we collect the mean pooled activations at each layer $\bar{\mathbf{h}}^{(1)}, \ldots, \bar{\mathbf{h}}^{(\ell)}$. and compute the cosine similarity between the pooled activation at each layer and that of the final layer:

$$\mathbf{x}_{\text{COSINE}} = \begin{bmatrix} \text{SIM}(\bar{\mathbf{h}}^{(1)}, \bar{\mathbf{h}}^{(\ell)}) & \ldots & \text{SIM}(\bar{\mathbf{h}}^{(\ell-1)}, \bar{\mathbf{h}}^{(\ell)}) \end{bmatrix}$$

This aggregation method yields $\ell - 1$ total features, *i.e.*, $\mathbf{x}_{\text{COSINE}} \in \mathbb{R}^{\ell-1}$. We chose COSINE as our similarity metric following previous work on aggregating embeddings to measure similarity, *e.g.*, BERTScore and Code-BERTScore (Zhang et al., 2020; Zhou et al., 2023). In practice, one could replace COSINE with any other measure. We use these $\ell - 1$ features to learn a confidence estimator. [9]

**PCA Transformed Activations** As mentioned earlier, the number of input features is still large, even when considering a single layer. Aggregating across layers is infeasible without dimensionality reduction, but collapsing each layer down to just one feature like our cosine approach removes a lot of information. We use Principal Components Analysis and compute the top $m$ components at each layer, leading to $m\ell$ features that we use to learn the estimator. We use 10 and 20 components in our experiments.

**3.2. ACUTE models**

We use these input features to train a simple classifier to predict whether a model generation is correct or not. Random forests are strong correctness predictors for tool-calling and question-answering tasks (Subramani et al., 2025a; Liu et al., 2025). [10] Traditionally, random forests require Platt scaling when used as confidence estimators as decision-trees favor extreme probabilities (Platt, 1999; Zadrozny & Elkan, 2001), but Shaker & Hüllermeier (2025) show that well-trained random forests do not need recalibration. We experiment without recalibration in our experiments unless stated otherwise. Hyperparameter details are in §4.

## 4. Experiments

We apply the ACUTE protocol on 6 models across 3 tasks and measure the degree to which calibration (via smECE) and utility (via EURO) improve over strong, efficient baselines. [11]

**Models** In our experiments, we evaluate 6 different models that come from 4 model families: gemma-3-4b-it, gemma-3-12b-it (Team et al., 2025), Qwen3-4B-Instruct-2507, Qwen3-14B (Yang et al., 2025), phi-4 (Abdin et al., 2024), and SmolLM3-3B (Bakouch et al., 2025). We apply the ACUTE protocol to each LLM separately and evaluate performance relative to strong baselines independently.

**Datasets & Tasks** To measure how ACUTE generalizes, we choose three distinct tasks: language understanding (MMLU; 5-shot in-context learning), tool-calling (API-Gen; zeroshot), and scientific document summarization (SC-ITLDR; zeroshot). MMLU is a multiple choice English-

---

[9]This estimator is a much more efficient version of the MICE estimators used to recalibrate tool-calling agents (Subramani et al., 2025a), see §C for more details.

[10]In early experiments, we tested other simple classifiers like logistic regression and support vector machines, and found that random forests were slightly more performant.

[11]See §C for a more detailed discussion.

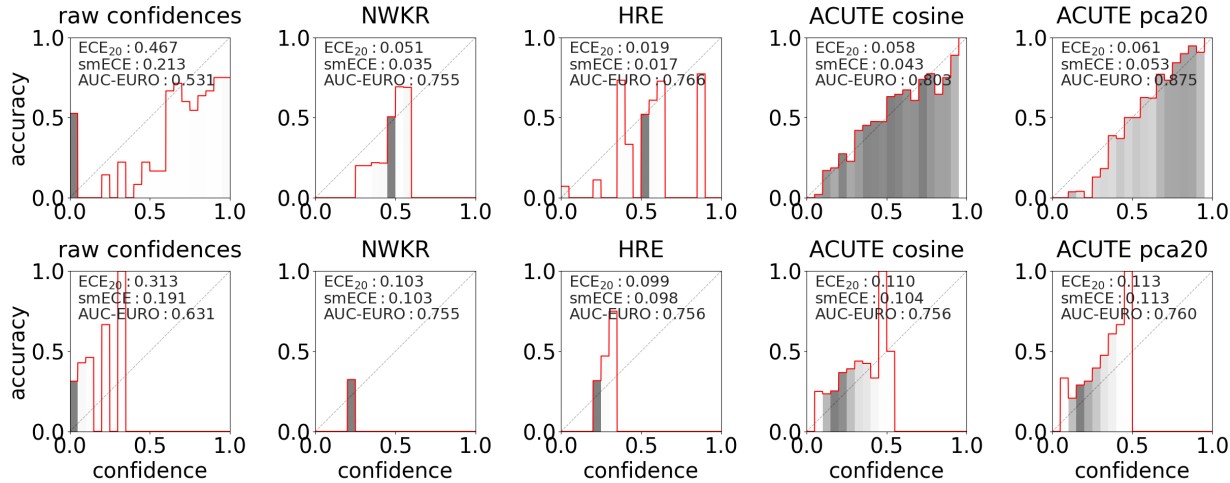

*Figure 3.* Here we show reliability diagrams for the gemma-3-12b-it model for APIGen (top row) and SCITLDR (bottom row) across 3 baseline estimators and two ACUTE estimators. Darker shading corresponds to higher density of examples in that confidence bin.

language benchmark with 57 distinct subtasks ranging from elementary school mathematics and sciences to college and post-graduate level medicine and law (Hendrycks et al., 2021). Crucially, these subtasks are all multiple choice, so LLMs generate a single token corresponding to the answer choice. APIGen is a tool-calling dataset where the model is given a description and set of APIs to potentially call (Liu et al., 2024). We instruct the LLM to generate a tool call or set of tool calls with API names and parameters in a parsable format. We experiment with the abstract only subtask of SC-ITLDR, where the model has to generate a short summary of approximately 25 words looking at just the abstract and title of the paper (Cachola et al., 2020). Both APIGen and SCITLDR require generating multi-token output and are evaluated using approximate exact match (APIGen) and rouge-L score (SCITLDR). Since we need binary labels for correctness, we use 0.3 as a threshold to stratify correct vs. incorrect generations for SCITLDR. See §J for an ablation over reasonable rouge-L threshold values. [12]

### 4.1. Baseline Confidence Estimators

**Raw Confidences:** Raw confidences are the default baseline that our language models emit. For an output sequence $y = w_1, \ldots, w_T$ of length $T$, the raw confidence is the joint probability that the language model $M$ assigns to $y$: $\hat{p} = \prod_{i=1}^{T} p_M(w_i|w_{<i})$. Computing this does not require any supervised training data or training an auxiliary confidence estimator as these are emitted directly from the language model. Raw confidence is used as an auxiliary feature for all of our ACUTE estimators because in early experiments, we saw a small performance gain when added. See §G for a more detailed discussion.

---

[12]See §E for system prompts and §F for data splits.

**Histogram Regression Estimator (HRE):** Our second baseline, HRE, is a standard recalibration method. We construct a histogram of confidences based on a training set with equal sized bins. To recalibrate a raw confidence estimate, we look up the bin associated with that estimate and return the average accuracy of all the samples in that bin. [13]

**Nadaraya-Watson Kernel Regression (NWKR):** For our third baseline, we use the Nadaraya-Watson kernel regression estimator (Nadaraya, 1964; Watson, 1964), which is the method used to calculate smECE. NWKR uses the raw confidence emitted by the language model and learns a reflected Gaussian kernel to map how over or under-confident a confidence estimate is. Formally, to map from a confidence estimate emitted by the language model $p_M$ to a recalibrated confidence $\hat{p}$, we look up the NWKR estimate for $p_M$ and return that estimate: $\hat{p} = NWKR(p_M)$, like with HRE.

**Other Posthoc Calibration Baselines:** We include three additional posthoc calibration baselines: Platt scaling (Platt, 1999), isotonic regression (Barlow et al., 1972), and beta calibration (Kull et al., 2017). All of these methods recalibrate a single probability estimate by fitting a function on held-out calibration set. Platt scaling fits a sigmoid, isotonic regression fits a step function with no defined functional form, and beta calibration fits a family of sigmoid functions via the beta cumulative distribution function.

### 4.2. ACUTE Estimators

We experiment with the different versions of the ACUTE estimators discussed in §3: early activations, middle activations, late activations, cosine, pca10, and pca20. As mentioned

---

[13]See Subramani et al. (2025a) for more details.

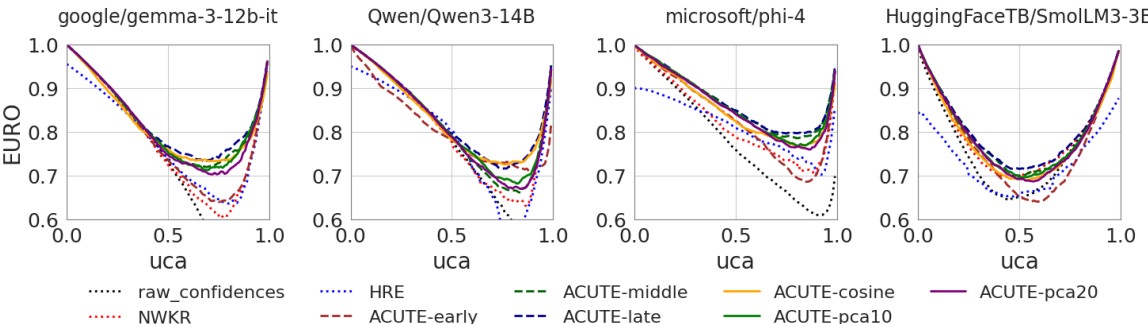

*Figure 4.* Results plotting EURO versus risk level ($u_{ca}$) for 4 of 6 LLMs (See Figure 9 for the full plot). Results are averaged across the 57 subtasks of MMLU. We include 3 baselines (raw confidences, HRE, and NWKR) along with our 6 configurations of ACUTE. All policies perform similarly at the lowest risk levels, but at high risk settings, the ACUTE estimators perform best.

earlier, ACUTE early, medium, and late are just the highest performing classifier using mean-pooled activations as input features over the first, second, and final third of the layers. ACUTE cosine uses the layer-wise cosine similarities with the final layer as input features to learn a confidence estimator. ACUTE pca10 and pca20 first PCA transform the activations at each layer to have 10 or 20 components and aggregate those across layers to produce input features. For all ACUTE models, we train a random forest classifier on these different features to predict whether the input query is correct or not. Our random forest uses 500 estimators, a max depth of 10, and does not restrict the number of features that can be used in each tree. Note that ACUTE estimators are *highly efficient*. For Qwen3-30B-A3B-Instruct-2507 on APIGen, generation took an average of 15.75 seconds per example on 2 L40S GPUs. The amortized time for inference on a trained random forest (ACUTE pca20) for an example is 0.00007 seconds, so the wall-clock overhead is *negligible* even if generation can be dramatically sped up.

## 5. Results & Analysis

### 5.1. Main Results

We present the results of evaluating the confidence estimators on MMLU, APIGen, and SCITLDR and evaluate both smECE and AUC-EURO in Table 1. We report 4 AUC-EURO values: low risk ($u_{ca} \in [0, \frac{1}{3})$), medium risk (med; $u_{ca} \in [\frac{1}{3}, \frac{2}{3})$), high risk ($u_{ca} \in [\frac{2}{3}, 1)$), and all ($u_{ca} \in [0, 1)$).

Raw confidences have high calibration error and low AUC-EURO values across the board; it is the worst performing estimator across all tasks. Inspecting the probabilities themselves (Figure 3), we observe rampant over-confidence in MMLU and under-confidence in APIGen and SCITLDR. Both HRE and NWKR improve upon the raw confidences by moving the probabilities into a much narrower range, whereby dramatically lowering smECE and increasing AUC-EURO. Due to this, both baselines lead to low utility confi-

dence estimates that make decision-making challenging, especially in high risk settings (see Figure 4 when $u_{ca} \in [0.4, 0.9]$). In contrast, our ACUTE estimators have greater probability mass spread, while maintaining the low calibration error of HRE and NWKR, and having higher AUC-EURO scores at all risk levels on the tasks.

Our best ACUTE systems have statistically significant improvements ($p < 0.05$) on AUC-EURO over the best baselines on MMLU and APIGen across all models and on SCITLDR for 6 out of the 7 models (excluding phi-4). We use permutation tests with 1000 resamples to test the difference of means of AUC-EURO across our ACUTE systems and NWKR. Additionally, the differences in smECE, where NWKR and HRE often outperform ACUTE, are not statistically significant ($p > 0.2$).

In Figure 3, we show reliability diagrams for the gemma3-12b-it model on APIGen and SCITLDR. Across tasks, HRE and NWKR bias for being a base rate estimator, whereas ACUTE systems have probability mass across the entire probability spectrum. ACUTE offers recalibration with decision-making utility, whereas HRE and NWKR bias for calibration alone. The best confidence estimators are ACUTE late and ACUTE pca20, having the highest AUC-EURO values across risk settings. Given SCITLDR is a challenging task, NWKR, HRE, and all ACUTE methods perform comparably metricwise, but baselines maintain degenerate behavior, nearly always predicting the base rate. ACUTE always outperforms baselines on AUC-EURO, while maintaining smECE.

### 5.2. Does ACUTE have strong sample complexity?

Here, we look at how sample efficient our methods are in comparison to the highly sample-efficient NWKR baseline. We look at effective training sizes $n_i = 25, 50, 100, 200, 500, 1000$ on the APIGen dataset. Recall the full training set has 1000 examples. For each effective training set size $n_i$, we randomly sample $n_i$ examples without replacement from the overall training set 20 times and train different

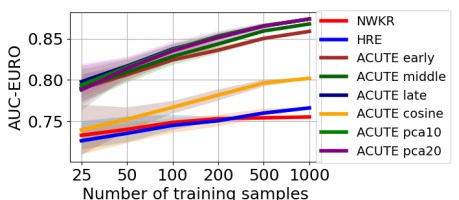
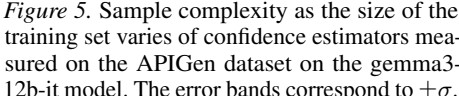

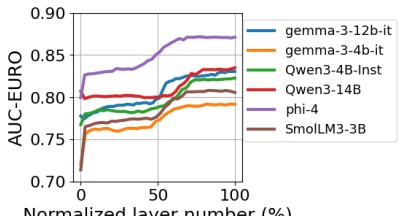

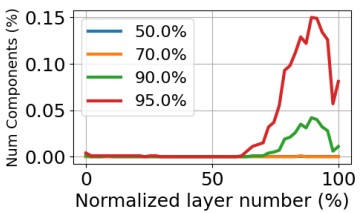

*Figure 5.* Sample complexity as the size of the training set varies of confidence estimators measured on the APIGen dataset on the gemma3-12b-it model. The error bands correspond to $\pm\sigma$.

*Figure 6.* AUC-EURO performance of our mean-pooled activation ACUTE estimators on all 6 LLMs on MMLU across layers.

*Figure 7.* Intrinsic dimension of the activations for gemma-3-12b-it on the APIGen test set.

estimators for each of these 20 samples. In Figure 5 we show the results of this for the gemma3-12b-it model with error bands corresponding to one standard deviation above and below the mean. We find that all ACUTE methods outperform both baselines at all effective training set sizes, with the activation-based and pca-based estimators (ACUTE early, middle, late, ACUTE pca10, and ACUTE pca20) performing better with just 25 training examples than either baseline (NWKR, HRE) on all 1000 training examples.

### 5.3. Which layers are best for ACUTE?

One of our configurations for ACUTE is using mean-pooled activations at a specific layer as input features to our random forest classifier. We dive deeper into this by looking at the performance of every layer across each of the 6 LLMs we tested on the MMLU benchmark, averaging results across all 57 tasks. In Figure 6, we find that layerwise performance is very low at the first layer, indicating that the subword embeddings alone lack a strong confidence estimation signal. Additionally, we find that there exist three phases that roughly correspond to the splits of early, middle, and late layers. Early corresponds to ∼40% of the layers, middle between 40 and 65%, and late the remaining. The early and late phases are flat, while the middle phase has a linearly increasing trend. These trends are consistent across LLMs, suggesting that models tend to encode information useful for confidence estimation similarly across layers.

### 5.4. Intrinsic Dimension Analysis

To better understand how complex the representations extracted from our language models are, we perform an intrinsic dimension analysis using PCA. [14] On the APIGen dataset, we look at how many components (as a percentage of the total) are needed to explain 50%, 70%, 90%, and 95% of the variance in our test set. Note that our test set has 1000 instances and our representation size for each of these

models is larger than 1000. Thus, trivially, 1000 components are sufficient to fully explain the variance in our data sample, so our maximum number of components is 1000. Figure 7 shows the results of inspecting the activations of gemma3-12b-it model on APIGen and analyzing its intrinsic dimension. We find that just a couple of components explain over 95% of the variance for the first 60% of the layers, but this increases at later layers perhaps providing better confidence estimation signal for our random forests. These trends are consistent with all LLMs we tested.

### 5.5. Does ACUTE generalize to larger MoE models?

We test how our ACUTE methods scale to Qwen3-30B-A3B-Instruct-2507 on both APIGen and SCITLDR (Yang et al., 2025). Table 2 shows good generalization; our earlier conclusions continue to hold for this model. In fact, these results suggest that ACUTE may have a larger confidence estimation boost for larger mixture-of-experts models.

### 5.6. Can posthoc calibration further improve ACUTE?

In other words, are the gains with posthoc calibration (*e.g.*, Platt scaling, isotonic regression, and beta calibration) additive with ACUTE? To test this, we posthoc calibrate the outputs of our ACUTE systems to reduce calibration error (smECE) on APIGen and SCITLDR. Figure 8 shows performance averaged across the 7 LLMs we tested. We find that when applying posthoc calibration to the base systems (denoted in black), AUC-EURO generally increases and calibration error (smECE) decreases, especially for SCITLDR. For APIGen on ACUTE cosine, posthoc calibration has negligible impact with isotonic regression actually lowering performance. Per model results are in Tables 12–19.

## 6. Related Work

**Calibration & Confidence Estimation:** Calibration has been studied in a variety of contexts often applied to binary classifiers. Often these works consider only a single predicted confidence and include Platt scaling (Platt, 1999), isotonic regression (Barlow et al., 1972), beta calibration (Kull

---

[14]This is common practice to understand the effective dimension or intrinsic dimension of representations (Li et al., 2018; Subramani et al., 2019; Li & Subramani, 2026a).

| | | APIGen | | | | | SCITLDR | | | |
| | | **AUC-EURO** (↑) | | | | | **AUC-EURO** (↑) | | | |
| estimator | smECE (↓) | low | med | high | all | smECE (↓) | low | med | high | all |
|---|---|---|---|---|---|---|---|---|---|---|
| Raw Conf | 0.261 | 0.465 | 0.471 | 0.594 | 0.510 | 0.146 | 0.413 | 0.760 | 0.939 | 0.701 |
| HRE | 0.028 | 0.941 | 0.799 | 0.718 | 0.820 | 0.088 | 0.637 | 0.763 | **0.940** | 0.779 |
| NWKR | 0.043 | 0.938 | 0.760 | 0.688 | 0.797 | 0.083 | 0.641 | 0.763 | **0.940** | 0.780 |
| Platt | 0.025 | 0.938 | 0.753 | 0.669 | 0.788 | **0.082** | 0.643 | 0.763 | **0.940** | 0.781 |
| Isotonic | **0.021** | 0.938 | 0.753 | 0.688 | 0.795 | **0.082** | 0.637 | 0.763 | **0.940** | 0.778 |
| Beta | 0.036 | 0.938 | 0.753 | 0.679 | 0.791 | 0.083 | 0.641 | 0.761 | **0.940** | 0.779 |
| ACUTE cosine | 0.031 | 0.946 | 0.817 | 0.757 | 0.841 | 0.089 | 0.634 | 0.760 | **0.940** | 0.776 |
| ACUTE pca10 | 0.025 | 0.946 | 0.834 | 0.813 | 0.865 | 0.092 | 0.637 | 0.762 | **0.940** | 0.778 |
| ACUTE pca20 | 0.027 | **0.947** | 0.843 | 0.812 | 0.868 | 0.092 | 0.653 | **0.765** | **0.940** | **0.785** |
| ACUTE early act | 0.047 | 0.942 | 0.825 | 0.784 | 0.851 | 0.089 | 0.652 | 0.763 | **0.940** | 0.784 |
| ACUTE mid act | 0.057 | 0.943 | 0.844 | 0.821 | 0.870 | 0.088 | 0.653 | 0.763 | **0.940** | 0.784 |
| ACUTE late act | 0.063 | 0.943 | **0.851** | **0.822** | **0.872** | 0.087 | **0.654** | 0.764 | **0.940** | 0.784 |

*Table 2.* Results on the APIGen test subset (left), and on the SCITLDR dev set (right) on Qwen3-30B-A3B-Instruct-2507. Lower smECE is better, while higher AUC-EURO is better. **Bold** indicates the best result and underline indicates the second best result in each column.

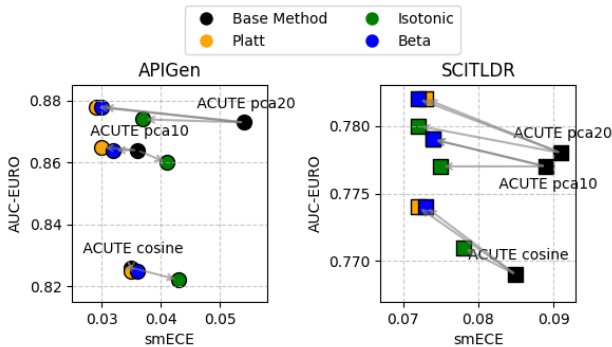

*Figure 8.* Effect of posthoc calibration with platt scaling, isotonic regression, and beta calibration on ACUTE systems on APIGEN (left) and SCITLDR (right). Posthoc calibration provides a reduction in calibration error without degrading AUC-EURO. Results here are averaged across the 7 LLMs evaluated in this paper.

**Model Internals:** Many works have studied the internals of language models to identify how linguistic information is encoded throughout the layers of a model (Hewitt & Manning, 2019; Jawahar et al., 2019; Tenney et al., 2019; Niu et al., 2022; He et al., 2024; Li & Subramani, 2026a; Acevedo et al., 2026). Language models can also be steered for exact generation (Subramani et al., 2019) and attribute-based steering (Subramani et al., 2022; Li et al., 2023). Others have found that there exist vectors that loosely correspond to functions or tasks (Hendel et al., 2023; Todd et al., 2024). Work combining mechanistic interpretability with calibration is rare: Beigi et al. (2024) use contrastive learning on model internals to improve trustworthiness, Subramani et al. (2025a) show that decoding from intermediate layers can improve calibration on tool-calling, and Liu et al. (2025) learn probes on activations to predict correctness in question-answering tasks. ACUTE operationalizes the activation spaces to build compute- and sample-efficient confidence estimators to improve trustworthiness.

# 7. Conclusion

We identify major problems with calibration metrics when used for decision-making and propose a new general, interpretable metric called **expected utility renormalized by the oracle** (EURO), which balances calibration and decision-making utility. EURO can reliably score confidence estimators, resolving some drawbacks of ECE. Also, we introduce a light-weight, compute- and sample-efficient activation-based confidence, utility, and trust estimation protocol (ACUTE) to recalibrate LLM outputs. Experiments on MCQA (MMLU), tool-calling (APIGen), and scientific document summarization (SCITLDR) show that incorporating the ACUTE protocol on 6 LLMs improves confidence estimation (EURO), while maintaining low calibration error.

et al., 2017), and histogram-based adaptive binning methods (Nobel, 1996). Additionally different language technology systems have had their calibration assessed including large language models (Jiang et al., 2021; Kadavath et al., 2022), and machine translation systems (Niculescu-Mizil & Caruana, 2005), while others leverage confidence estimates to ask for confirmation (Stengel-Eskin & Van Durme, 2023b), and provide an estimate of trust to users (Stengel-Eskin et al., 2024). Finally, as language model capabilities have improved, linguistic calibration, *i.e.*, asking the language model of its confidence, has become popular and has mixed utility (Mielke et al., 2022; Band et al., 2024). Prior work focus on single-token, classification-like settings, ACUTE, on the other hand, works with both single- and multi-token generation settings and offers a generalized activation-based calibration solution.

## Impact Statement

In this paper, we present EURO and ACUTE. EURO is a metric to measure how well calibrated and how useful a confidence estimator is for decision-making. This gives users an alternative metric to measure how well their systems work across a variety of risk-levels. ACUTE provides a light-weight, compute- and sample-efficient method to directly recalibrate LLM output probabilities. Even though ACUTE requires access to model internals, the developers of a language model will always have access to its internals. In this competitive landscape, LLM developers are incentivized to improve user trust in their system, so incorporating ACUTE would be beneficial to that end.

Since better calibration improves decision-making for users, both of our contributions can help increase trust in language model outputs, either by helping to provide more reliable confidence estimates (through ACUTE) or by evaluating an already deployed confidence estimator (through EURO).

We believe that as LLMs become increasingly embedded in everyday technology, reliability and trust is paramount. We believe our work takes a small step to reiterate the need for better metrics and methods to appropriately adjudicate trust. All that being said, our decision theoretic framework is not a replacement for careful data analysis and system design.

## Acknowledgments

We would like to thank Nivedita Suresh for feedback on §2. Additionally, we would like to thank the anonymous reviewers for numerous suggestions in the paper, including adding posthoc calibration methods as well as testing generalization to large MoE style models, which we did with Qwen3-30B-A3B-Instruct-2507.

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

# A. EURO derivations

Here we present some of the derivations from §2.

## A.1. Bayes Optimality of the MBR Threshold

Suppose a confidence estimator emits a probability $\hat{p}$ for an example. From our vantage point, we have no other information than just this probability $\hat{p}$. We expect that the probability that the example is correct is $\hat{p}$ and thus the probability that the example is incorrect is $1 - \hat{p}$. For this task, lets assume we get the following rewards for true positives, true negatives, false positives, and false negatives: $R_{tp}$, $R_{tn}$, $R_{fp}$, and $R_{fn}$. The reward associated with predicting the positive class (trusting the estimator) is:

$$\hat{p} * R_{tp} + (1 - \hat{p}) * R_{fp} \tag{6}$$

The reward associated with predicting the negative class (abstaining from trusting the estimator) is:

$$(1 - \hat{p}) * R_{tn} + (\hat{p}) * R_{fn} \tag{7}$$

Given this, we should trust the estimator (predict the positive class) if and only if the reward for this is greater than the reward for abstaining (predicting the negative class):

$$\hat{p}(R_{tp}) + (1 - \hat{p})R_{fp} > (1 - \hat{p})R_{tn} + \hat{p}(R_{fn})$$
$$\hat{p}(R_{tp} - R_{fn}) > (1 - \hat{p})(R_{tn} - R_{fp})$$
$$\hat{p}(R_{tp} - R_{fn}) > \hat{p}(R_{fp} - R_{tn}) + R_{tn} - R_{fp}$$
$$\hat{p}(R_{tp} - R_{fn} - R_{fp} + R_{tn}) > R_{tn} - R_{fp}$$
$$\hat{p} > \frac{R_{tn} - R_{fp}}{(R_{tp} - R_{fn} - R_{fp} + R_{tn})}$$

We rearrange this to get the Minimum Bayes Risk threshold $\tau$ from equation 1:

$$\hat{p} > \tau \stackrel{\text{def}}{=} \frac{R_{tn} - R_{fp}}{(R_{tp} - R_{fn}) + (R_{tn} - R_{fp})} \tag{8}$$

.

## A.2. Recomputing the MBR Threshold

Recall that $U_{ct} = R_{tp} - R_{fn}$ and $U_{ca} = R_{tn} - R_{fp}$. Substituting the definitions of $U_{ct}$, $U_{ca}$, $u_{ct}$, and $u_{ca}$ into equation 1 yields:

$$\hat{p} > \tau \stackrel{\text{def}}{=} \frac{R_{tn} - R_{fp}}{(R_{tp} - R_{fn}) + (R_{tn} - R_{fp})} = \frac{U_{ca}}{U_{ct} + U_{ca}} = u_{ca}$$

## A.3. Calculating the Relative Utility

Here we show the proof for equation 4. Recall that we are aiming to calculate the relative utility between two confidence estimators $C_1$ and $C_2$. The rewards ($R_{tp}$, $R_{tn}$, $R_{fp}$, and $R_{fn}$) and language model generations ($\hat{y}_1, \ldots \hat{y}_n$) are fixed. The confusion matrices are the only differences between the systems, so the only thing that varies are the counts of true positives, true negatives, false positives, and false negatives between the two systems based on equation 3. We use the total reward calculation $\mathbb{U} = R_{tp} * N_{tp} + R_{tn} * N_{tn} + R_{fp} * N_{fp} + R_{fn} * N_{fn}$ to compute the relative utility between two systems. This is precisely the difference in total utility: $\mathbb{RU}_{C_1, C_2} = \mathbb{U}_{C_1} - \mathbb{U}_{C_2}$. Remember that our generations are fixed. This ensures that the number of correct and incorrect samples are preserved: $N_{tp, C_1} + N_{fn, C_1} = N_{tp, C_2} + N_{fn, C_2}$

and $N_{tn,C_1} + N_{fp,C_1} = N_{tn,C_2} + N_{fp,C_2}$. Rearranging this shows that $N_{tp,C_1} - N_{tp,C_2} = -N_{fn,C_1} + N_{fn,C_2}$ and $N_{tn,C_1} - N_{tn,C_2} = -N_{fp,C_1} + N_{fp,C_2}$.

$$\mathbb{RU}_{C_1,C_2} = \mathbb{U}_{C_1} - \mathbb{U}_{C_2} = \begin{bmatrix} N_{tp,C_1} - N_{tp,C_2} \\ N_{tn,C_1} - N_{tn,C_2} \\ N_{fp,C_1} - N_{fp,C_2} \\ N_{fn,C_1} - N_{fn,C_2} \end{bmatrix} \cdot \begin{bmatrix} R_{tp} \\ R_{tn} \\ R_{fp} \\ R_{fn} \end{bmatrix}$$

$$\mathbb{RU}_{C_1,C_2} = \begin{bmatrix} N_{tp,C_1} - N_{tp,C_2} \\ N_{tn,C_1} - N_{tn,C_2} \\ -N_{tn,C_1} + N_{tn,C_2} \\ -N_{tp,C_1} + N_{tp,C_2} \end{bmatrix} \cdot \begin{bmatrix} R_{tp} \\ R_{tn} \\ R_{fp} \\ R_{fn} \end{bmatrix}$$

$$\mathbb{RU}_{C_1,C_2} = \begin{bmatrix} (R_{tp} - R_{fn}) \\ (R_{tn} - R_{fp}) \end{bmatrix} \cdot \begin{bmatrix} (N_{tp,C_1} - N_{tp,C_2}) \\ (N_{tn,C_1} - N_{tn,C_2}) \end{bmatrix}$$

$$\mathbb{RU}_{C_1,C_2} = \begin{bmatrix} U_{ct} \\ U_{ca} \end{bmatrix} \cdot \begin{bmatrix} (N_{tp,C_1} - N_{tp,C_2}) \\ (N_{tn,C_1} - N_{tn,C_2}) \end{bmatrix}$$

We can write this in terms of $u_{ct}$ and $u_{ca}$, using equation 2:

$$\mathbb{RU}_{C_1,C_2} = \frac{1}{U_{ct} + U_{ca}} \cdot \begin{bmatrix} u_{ct} \\ u_{ca} \end{bmatrix} \cdot \begin{bmatrix} (N_{tp,C_1} - N_{tp,C_2}) \\ (N_{tn,C_1} - N_{tn,C_2}) \end{bmatrix} \tag{9}$$

### A.4. Renormalizing with respect to the Oracle and Anti-Oracle Derivation

Recall that we want our relative utilities to be scaled with respect to the Oracle ($O$) and Anti-Oracle ($AO$) policies, with $\text{EURO}_O = 1$ and $\text{EURO}_{AO} = 0$. For a confidence estimator $C$, EURO is calculated as follows:

$$\text{EURO}_C(u_{ca}) = \frac{\mathbb{RU}_{C,AO}}{\mathbb{RU}_{O,AO}} \tag{10}$$

Notice that the $\frac{1}{U_{ct}+U_{ca}}$ is a constant in both relative utilities, so we can cancel those out.

$$\text{EURO}_C(u_{ca}) = \begin{bmatrix} u_{ct} \\ u_{ca} \end{bmatrix} \cdot \frac{\begin{bmatrix} (N_{tp,C} - N_{tp,AO}) \\ (N_{tn,C} - N_{tn,AO}) \end{bmatrix}}{\begin{bmatrix} (N_{tp,O} - N_{tp,AO}) \\ (N_{tn,O} - N_{tn,AO}) \end{bmatrix}}$$

The Anti-Oracle policy has no true positives or true negatives ($N_{tp,AO} = N_{tn,AO} = 0$). Also recall that $u_{ct} = 1 - u_{ca}$.

$$\text{EURO}_C(u_{ca}) = \frac{u_{ct} \cdot N_{tp,C} + u_{ca} \cdot N_{tn,C}}{u_{ct} \cdot N_{tp,O} + u_{ca} \cdot N_{tn,O}} = \frac{N_{tp,C} + u_{ca} \cdot (N_{tn,C} - N_{tp,C})}{N_{tp,O} + u_{ca} \cdot (N_{tn,O} - N_{tp,O})}$$

## B. How EURO relates to other measures

EURO is unique in that it combines calibration with decision-making utility together in one simple metric. As mentioned earlier, calibration error metrics have major drawbacks that make decision-making challenging. One of these drawbacks is the lack of incorporating the risk-level or the costs (and rewards) of true positives, false positives, true negatives, and false negatives into account. An estimator that emits the same sets of probabilities for examples should be adjudicated differently for a high risk task versus a low risk one. For a task with 0 risk, the right thing to do is to always trust your output regardless of whether it is correct. As a result, any estimator that generates a nonzero probability would be scored equally. Both calibration error metrics (ECE, smECE) lack this. Brier score lacks this too, but has the guarantee that an objective function that minimizes it, will be Bayes-optimal.

Precision, recall, sensitivity, and specificity deal with the confusion matrix directly. The area under the precision recall curve (AUPRC) and the area under the receiver operating characteristic curve (AUROC) both involve summarizing the

confusion matrix trading off these quantities. In general, AUROC treats both classes as equal, so it is invariant to risk level. AUPRC is not symmetric and is often used when there are imbalanced classes. In a high (or low) risk scenario the rewards for the classes are imbalanced and thus AUPRC could be used. However, both AUROC and AUPRC are reward-blind, so a MBR type of formulation needs to be used. EURO balances calibration and decision-making utility, while being able to appropriately score a confidence estimator across risk levels, taking into account calibration.

## C. Confidence Estimators can be Computationally Infeasible

Confidence estimators need to be light-weight and computationally-efficient to be useful in a variety of settings and actually be used in practice for language model generation. Most methods for confidence estimation and model recalibration are expensive, nearly all applications ignore confidence estimation. The ones that attempt recalibration rely on raw confidences or adhoc linguistic calibration on a small set of outputs.

Previous work on calibration recalibrates in an inefficient manner. Subramani et al. (2025a) propose MICE, which, to our knowledge, is the only model internals based confidence estimation work that leverages hidden states to recalibrate language model outputs in a single shot. MICE requires logit lens decoding from each intermediate layer, which by itself is expensive, requiring additional matrix multiplies with the unembedding matrix for each layer. Additionally, they compute the BERTSCORE between the final output generation and each intermediate generation and requires using a large auxiliary language model (DeBERTa-xlarge-mnli) to extract these BERTSCORE features before training their classifier. This is computationally infeasible. Even if the confidence estimator has been trained, each new inference query would require an additional forward passes of the BERTSCORE model equal to the number of layers of the model along with the logit lens decoding overhead. Assuming the underlying language model is the same size as DeBERTa-xlarge-mnli, each query takes 40-50x longer than the simple generation.

Our ACUTE cosine method approximates MICE. We remove the reliance on the auxiliary model to compute BERTSCORE by looking at the model's activations and comparing those with other layers. This also removes the reliance on intermediate decoding, improving efficiency by at least 40x, and adding negligible overhead for confidence estimation.

## D. Connection to Selective Prediction

From our understanding, a selective prediction system can decide to output a class label or abstain from it and is governed by a selective function. In our setting, our selective function is a combination of the confidence estimator and the minimum Bayes risk-based optimal policy. The selective classifier tries to minimize the selective risk at a given coverage value. In our setting, we define the risk level according to the normalized net utility of correctly abstaining ($u_{ca}$) based on the Bayes-optimal policy, and evaluate EURO according to that risk level. Given this, the risk-coverage curve defined in selective prediction literature and our $u_{ca}$ vs. EURO curve are different.

## E. Prompts used for tasks

### E.1. MMLU

```
"Answer the following multiple choice question by giving the most appropriate response.
    Answer with a single letter: either 'A', 'B', 'C', or 'D'. Even though the answer
    choices may have a period at the end, answer with just the letter. The following are
    multiple choice questions (with answers) about {CURR_SUBJECT} Question: {ICL EXAMPLE1}
     Answer: {ICL ANSWER1} ... Question: {ICL EXAMPLE 5} Answer: {ICL ANSWER5} Now answer
    the following multiple choice question with a single letter. Question: {QUESTION}
    Answer: "
```

*Listing 1.* System Prompt for all models for MMLU

### E.2. APIGen

```
"You are an agent that specializes in function calling. You are given a list of tools,
    each with a name, description, and arguments. Each of the arguments may have a name,
    description, type, and a default value that the argument takes. You will be given a
    query. Please answer the query using the provided tools. Format your final answer as
    list of function calls in a json-like format without any linebreaks. Make sure the
```

```
arugments for the function-calling are called arguments, not parameters. Here are the
    list of tools available: {CURRENT_EXAMPLES_AVAILABLE_TOOLS}. {INPUT_QUERY}"
```

*Listing 2.* System Prompt for all models for APIGen

## E.3. SCITLDR

```
"You are a helpful assistant who is an expert in summarizing scientific papers. Given the
    title and abstract of the paper in the following format TITLE: <title> ABSTRACT: <
    abstract> SUMMARY: , write a short summary of the abstract, which will go after the
    SUMMARY: in the prompt. Summaries must be no more than one sentence, but this sentence
     can be longer than an average sentence. Most summaries are between 15 and 25 words.
    TITLE: {CURR_EXAMPLE_TITLE} ABSTRACT: {CURR_EXAMPLE_ABSTRACT} SUMMARY: "
```

*Listing 3.* System Prompt for all models for SCITLDR

## F. Dataset Information

For MMLU, for each task we split into training and evaluation sets. If the task had less than 200 examples, we used 50 for training, less than 400 examples, 100 for training, and otherwise 200 for training. All other examples were put in the evaluation set. For APIGen we randomly assigned 1000 examples for training and 1000 for evaluation. For SCITLDR, we use the training set (1992 samples) for training and the development set (618 samples) for evaluation. When including the post-hoc calibration, we split each training set into 80% training and 20% calibration.

## G. Confidence Feature Ablation

Here we show results from early experiments when adding raw confidence as a feature to our ACUTE systems. These are averaged across 6 LLMs we test (all models not including Qwen3-30B-A3B-Instruct-2507 and result in a negligibly positive differences.

| Estimator | Dataset | $\Delta$**smECE** ($\downarrow$) | $\Delta$AUC-EURO ($\uparrow$) |
|---|---|---|---|
| ACUTE pca10 | MMLU (all) | 0.0 | 0.002 |
| ACUTE pca20 | MMLU (all) | 0.0 | 0.001 |
| ACUTE pca10 | APIGen | 0.0 | 0.0 |
| ACUTE pca20 | APIGen | 0.002 | 0.0 |
| ACUTE pca10 | SCITLDR | 0 | 0.0 |
| ACUTE pca20 | SCITLDR | 0.001 | 0.0 |

*Table 3.* Here we show the difference in smECE and AUC-EURO when adding the confidence feature for ACUTE pca10 and ACUTE pca20 across all three datasets. Deltas are very small indicating that the feature is not vital to include.

## H. Noise Ablation

To test whether any multi-variate vectors help with confidence estimation, we construct a noise matrix by sampling from a multi-variate gaussian parametrized by each feature's mean and variance. We interpolate between the original features and the noise matrix on both training and test sets and evaluate ACUTE pca20 via smECE and AUC-EURO averaged across all LLMs on APIGen. N% noise corresponds to $N * X_{noise} + (1 - N) * X_{features}$ We include the base rate estimator (one that just predicts training set accuracy for all inputs). Results hint that the activations contain relevant signals for confidence estimation and that performance degrades to the base rate estimator with more noise.

## I. MMLU results across LLMs

In Figure 9, we show the full plot of results across all 6 LLMs. We find that trends are very consistent and show that ACUTE systems perform best at high risk levels and comparably to baselines at lower risk levels.

| Noise Level | smECE ($\downarrow$) | AUC-EURO ($\uparrow$) |
|---|---|---|
| 0% | 0.05 | **0.873** |
| 25% | 0.06 | 0.867 |
| 50% | 0.06 | 0.850 |
| 75% | 0.05 | 0.814 |
| 100% | 0.05 | 0.778 |
| Base Rate Estimator | 0.001 | 0.777 |

*Table 4.* Here we show the smECE and AUC-EURO when adding different magnitudes of Gaussian noise to the input data for the ACUTE pca20 estimator on APIGen. Intuitively, increased noise degrades performance on decision-making utility.

| | | Threshold = 0.2 | | | | | Threshold = 0.3 | | | | | Threshold = 0.4 | | | |
|---|---|---|---|---|---|---|---|---|---|---|---|---|---|---|---|
| | | AUC-EURO ($\uparrow$) | | | | | AUC-EURO ($\uparrow$) | | | | | AUC-EURO ($\uparrow$) | | | |
| estimator | smECE ($\downarrow$) | low | med | high | all | smECE ($\downarrow$) | low | med | high | all | smECE ($\downarrow$) | low | med | high | all |
| Raw Conf | 0.39 | 0.10 | 0.22 | 0.60 | 0.30 | 0.19 | 0.32 | 0.67 | 0.91 | 0.63 | 0.04 | 0.67 | 0.92 | 0.98 | 0.86 |
| HRE | 0.20 | 0.94 | 0.69 | 0.60 | 0.75 | 0.10 | 0.69 | 0.67 | 0.91 | 0.76 | 0.02 | 0.74 | 0.92 | 0.98 | 0.88 |
| NWKR | 0.20 | 0.95 | 0.69 | 0.60 | 0.75 | 0.10 | 0.69 | 0.67 | 0.91 | 0.75 | 0.02 | 0.74 | 0.92 | 0.98 | 0.88 |
| Platt | 0.20 | 0.95 | 0.70 | 0.60 | 0.75 | 0.10 | 0.68 | 0.67 | 0.91 | 0.75 | 0.02 | 0.73 | 0.92 | 0.98 | 0.88 |
| Isotonic | 0.20 | 0.95 | 0.70 | 0.60 | 0.75 | 0.10 | 0.69 | 0.67 | 0.91 | 0.76 | 0.02 | 0.74 | 0.92 | 0.98 | 0.88 |
| Beta | 0.20 | 0.95 | 0.69 | 0.60 | 0.75 | 0.10 | 0.69 | 0.67 | 0.91 | 0.75 | 0.02 | 0.73 | 0.92 | 0.98 | 0.88 |
| ACUTE cosine | 0.20 | 0.95 | 0.68 | 0.61 | 0.75 | 0.10 | 0.69 | 0.67 | 0.91 | 0.75 | 0.03 | 0.72 | 0.92 | 0.98 | 0.87 |
| ACUTE pca10 | 0.20 | 0.95 | 0.68 | 0.61 | 0.75 | 0.11 | 0.70 | 0.68 | 0.91 | 0.76 | 0.03 | 0.72 | 0.92 | 0.98 | 0.87 |
| ACUTE pca20 | 0.20 | 0.95 | 0.70 | 0.61 | 0.75 | 0.11 | 0.70 | 0.67 | 0.91 | 0.76 | 0.03 | 0.72 | 0.92 | 0.98 | 0.87 |
| ACUTE early act | 0.20 | 0.95 | 0.71 | 0.60 | 0.75 | 0.11 | 0.70 | 0.67 | 0.91 | 0.76 | 0.02 | 0.74 | 0.92 | 0.98 | 0.88 |
| ACUTE mid act | 0.20 | 0.95 | 0.71 | 0.60 | 0.76 | 0.10 | 0.70 | 0.67 | 0.91 | 0.76 | 0.02 | 0.74 | 0.92 | 0.98 | 0.88 |
| ACUTE late act | 0.20 | 0.95 | 0.71 | 0.60 | 0.76 | 0.10 | 0.69 | 0.67 | 0.91 | 0.76 | 0.02 | 0.73 | 0.92 | 0.98 | 0.88 |

*Table 5.* Results on the SCITLDR dev set evaluated at different Rouge-L thresholds to test threshold sensitivity on gemma3-12b-it. Lower smECE is better, while higher AUC-EURO is better.

## J. SCITLDR Threshold Analysis

In Tables 5–11, we show the per model results for varying the Rouge-L threshold to decide whether an example is correct or not across the LLMs we tested. Note the mean accuracies (correct vs. incorrect on the language model output) across these LLMs for different rouge-L thresholds are approximately 60% for 0.2, 22% for 0.3, and 6% for 0.4 with low variances across models.

## K. Per Model Results

We show per model results on APIGen and SCITLDR including posthoc calibration of ACUTE systems in Tables 12–19. Generally, we find that ACUTE systems outperform baselines and that posthoc calibration lowers calibration error while maintaining or sometimes improving AUC-EURO for most models.

## L. Additional Limitations

Our work has several limitations. For multi-token generation settings, many tokens are formatting-based tokens and perhaps should not be fully considered in the joint probability when computing raw confidence, but we found it challenging to pinpoint these, so we defaulted to including all of them. Secondly, for many of these tasks more stochasticity in the generation process may offer stronger performance, especially given that different versions of truncation sampling drastically outperform greedy decoding in most text generation settings. However, for replicability and to remove sources of stochasticity, we elected for greedy decoding. Another limitation could be the requirement to have unfettered access to model internals, although for every model someone, often the model developer and anyone in that sub-organization, have access to model internals.

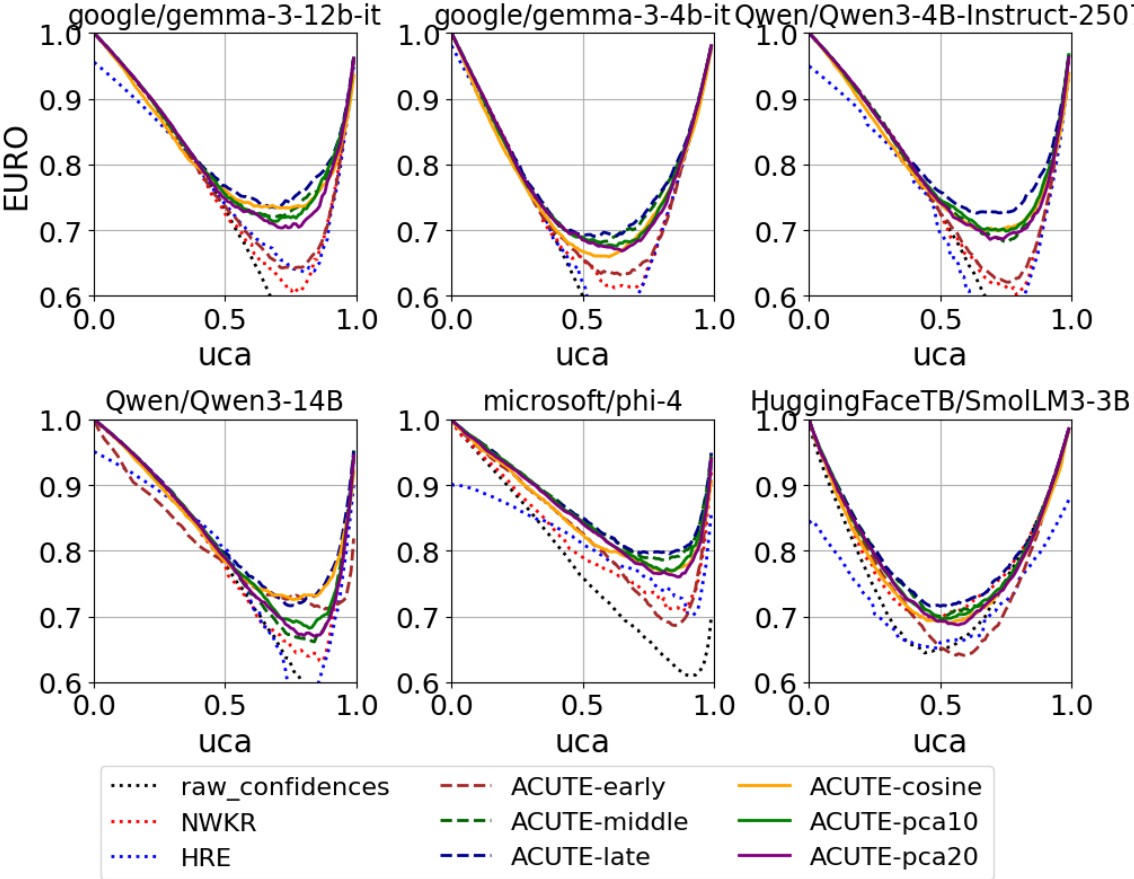

Figure 9. Results plotting EURO versus risk level ($u_{ca}$) for 6 LLMs. Performances are averaged across the 57 subtasks of MMLU. We include 3 baselines (raw confidences, HRE, and NWKR) along with our 6 configurations of ACUTE. All policies perform similarly at the lowest risk levels, but at high risk settings, the ACUTE estimators perform best.

| | **Threshold = 0.2** | | | | | **Threshold = 0.3** | | | | | **Threshold = 0.4** | | | | |
|---|---|---|---|---|---|---|---|---|---|---|---|---|---|---|---|
| | | **AUC-EURO** (↑) | | | | | **AUC-EURO** (↑) | | | | | **AUC-EURO** (↑) | | | |
| **estimator** | **smECE** (↓) | low | med | high | all | **smECE** (↓) | low | med | high | all | **smECE** (↓) | low | med | high | all |
| Raw Conf | 0.38 | 0.11 | 0.26 | 0.64 | 0.33 | 0.17 | 0.36 | 0.71 | 0.92 | 0.66 | 0.03 | 0.67 | 0.93 | 0.98 | 0.86 |
| HRE | 0.20 | 0.94 | 0.63 | 0.64 | 0.74 | 0.08 | 0.67 | 0.71 | 0.92 | 0.76 | 0.01 | 0.74 | 0.93 | 0.98 | 0.88 |
| NWKR | 0.19 | 0.94 | 0.65 | 0.64 | 0.74 | 0.08 | 0.67 | 0.71 | 0.92 | 0.77 | 0.01 | 0.74 | 0.93 | 0.98 | 0.88 |
| Platt | 0.19 | 0.94 | 0.64 | 0.64 | 0.74 | 0.07 | 0.67 | 0.71 | 0.92 | 0.77 | 0.01 | 0.74 | 0.93 | 0.98 | 0.88 |
| Isotonic | 0.20 | 0.94 | 0.63 | 0.64 | 0.74 | 0.08 | 0.68 | 0.71 | 0.92 | 0.77 | 0.01 | 0.74 | 0.93 | 0.98 | 0.88 |
| Beta | 0.20 | 0.94 | 0.64 | 0.64 | 0.74 | 0.07 | 0.67 | 0.71 | 0.92 | 0.77 | 0.01 | 0.74 | 0.93 | 0.98 | 0.88 |
| ACUTE cosine | 0.19 | 0.93 | 0.64 | 0.64 | 0.74 | 0.08 | 0.65 | 0.71 | 0.92 | 0.76 | 0.03 | 0.72 | 0.92 | 0.98 | 0.88 |
| ACUTE pca10 | 0.20 | 0.93 | 0.64 | 0.65 | 0.74 | 0.09 | 0.68 | 0.72 | 0.92 | 0.77 | 0.03 | 0.73 | 0.93 | 0.98 | 0.88 |
| ACUTE pca20 | 0.20 | 0.94 | 0.64 | 0.65 | 0.74 | 0.09 | 0.69 | 0.71 | 0.92 | 0.77 | 0.02 | 0.74 | 0.93 | 0.98 | 0.88 |
| ACUTE early act | 0.19 | 0.94 | 0.67 | 0.64 | 0.75 | 0.08 | 0.70 | 0.71 | 0.92 | 0.78 | 0.02 | 0.75 | 0.93 | 0.98 | 0.88 |
| ACUTE mid act | 0.19 | 0.94 | 0.67 | 0.64 | 0.75 | 0.08 | 0.69 | 0.71 | 0.92 | 0.77 | 0.02 | 0.74 | 0.93 | 0.98 | 0.88 |
| ACUTE late act | 0.19 | 0.94 | 0.66 | 0.64 | 0.75 | 0.08 | 0.69 | 0.71 | 0.92 | 0.77 | 0.02 | 0.74 | 0.93 | 0.98 | 0.88 |

Table 6. Results on the SCITLDR dev set evaluated at different Rouge-L thresholds to test threshold sensitivity on gemma3-4b-it. Lower smECE is better, while higher AUC-EURO is better.

| | | Threshold = 0.2 | | | | | Threshold = 0.3 | | | | | Threshold = 0.4 | | | |
|---|---|---|---|---|---|---|---|---|---|---|---|---|---|---|---|
| | | AUC-EURO (↑) | | | | | AUC-EURO (↑) | | | | | AUC-EURO (↑) | | | |
| estimator | smECE (↓) | low | med | high | all | smECE (↓) | low | med | high | all | smECE (↓) | low | med | high | all |
| Raw Conf | 0.37 | 0.16 | 0.27 | 0.65 | 0.36 | 0.16 | 0.38 | 0.71 | 0.92 | 0.67 | 0.03 | 0.74 | 0.94 | 0.99 | 0.89 |
| HRE | 0.18 | 0.93 | 0.63 | 0.66 | 0.74 | 0.07 | 0.66 | 0.72 | 0.92 | 0.77 | 0.01 | 0.77 | 0.95 | 0.99 | 0.90 |
| NWKR | 0.18 | 0.93 | 0.64 | 0.65 | 0.74 | 0.08 | 0.67 | 0.71 | 0.92 | 0.77 | 0.01 | 0.78 | 0.95 | 0.99 | 0.90 |
| Platt | 0.18 | 0.93 | 0.64 | 0.65 | 0.74 | 0.07 | 0.67 | 0.71 | 0.92 | 0.77 | 0.01 | 0.78 | 0.95 | 0.99 | 0.90 |
| Isotonic | 0.18 | 0.93 | 0.65 | 0.65 | 0.75 | 0.07 | 0.68 | 0.71 | 0.92 | 0.77 | 0.01 | 0.78 | 0.95 | 0.99 | 0.90 |
| Beta | 0.18 | 0.93 | 0.65 | 0.65 | 0.75 | 0.07 | 0.67 | 0.71 | 0.92 | 0.77 | 0.01 | 0.78 | 0.95 | 0.99 | 0.90 |
| ACUTE cosine | 0.18 | 0.93 | 0.64 | 0.66 | 0.74 | 0.08 | 0.68 | 0.71 | 0.92 | 0.77 | 0.03 | 0.76 | 0.95 | 0.99 | 0.90 |
| ACUTE pca10 | 0.18 | 0.93 | 0.65 | 0.66 | 0.75 | 0.08 | 0.70 | 0.71 | 0.92 | 0.78 | 0.02 | 0.78 | 0.95 | 0.99 | 0.90 |
| ACUTE pca20 | 0.18 | 0.93 | 0.67 | 0.66 | 0.76 | 0.08 | 0.70 | 0.72 | 0.92 | 0.78 | 0.02 | 0.78 | 0.95 | 0.99 | 0.90 |
| ACUTE early act | 0.17 | 0.93 | 0.67 | 0.66 | 0.76 | 0.08 | 0.69 | 0.71 | 0.92 | 0.77 | 0.01 | 0.79 | 0.95 | 0.99 | 0.91 |
| ACUTE mid act | 0.17 | 0.93 | 0.67 | 0.66 | 0.76 | 0.08 | 0.69 | 0.71 | 0.92 | 0.77 | 0.01 | 0.79 | 0.95 | 0.99 | 0.91 |
| ACUTE late act | 0.18 | 0.93 | 0.67 | 0.66 | 0.75 | 0.08 | 0.68 | 0.71 | 0.92 | 0.77 | 0.01 | 0.78 | 0.95 | 0.99 | 0.91 |

*Table 7.* Results on the SCITLDR dev set evaluated at different Rouge-L thresholds to test threshold sensitivity on Qwen3-4B-Instruct-2507. Lower smECE is better, while higher AUC-EURO is better.

| | | Threshold = 0.2 | | | | | Threshold = 0.3 | | | | | Threshold = 0.4 | | | |
|---|---|---|---|---|---|---|---|---|---|---|---|---|---|---|---|
| | | AUC-EURO (↑) | | | | | AUC-EURO (↑) | | | | | AUC-EURO (↑) | | | |
| estimator | smECE (↓) | low | med | high | all | smECE (↓) | low | med | high | all | smECE (↓) | low | med | high | all |
| Raw Conf | 0.38 | 0.12 | 0.24 | 0.62 | 0.32 | 0.16 | 0.34 | 0.69 | 0.91 | 0.65 | 0.05 | 0.61 | 0.90 | 0.98 | 0.83 |
| HRE | 0.19 | 0.94 | 0.68 | 0.62 | 0.75 | 0.09 | 0.68 | 0.69 | 0.91 | 0.76 | 0.03 | 0.71 | 0.90 | 0.98 | 0.86 |
| NWKR | 0.19 | 0.94 | 0.67 | 0.62 | 0.75 | 0.11 | 0.67 | 0.69 | 0.91 | 0.76 | 0.04 | 0.70 | 0.90 | 0.98 | 0.86 |
| Platt | 0.19 | 0.94 | 0.68 | 0.62 | 0.75 | 0.09 | 0.68 | 0.69 | 0.91 | 0.76 | 0.03 | 0.70 | 0.90 | 0.98 | 0.86 |
| Isotonic | 0.19 | 0.94 | 0.68 | 0.62 | 0.75 | 0.10 | 0.67 | 0.69 | 0.91 | 0.76 | 0.03 | 0.70 | 0.90 | 0.98 | 0.86 |
| Beta | 0.19 | 0.94 | 0.68 | 0.62 | 0.75 | 0.09 | 0.68 | 0.69 | 0.91 | 0.76 | 0.03 | 0.70 | 0.90 | 0.98 | 0.86 |
| ACUTE cosine | 0.19 | 0.94 | 0.67 | 0.62 | 0.75 | 0.09 | 0.68 | 0.69 | 0.91 | 0.76 | 0.05 | 0.70 | 0.90 | 0.98 | 0.86 |
| ACUTE pca10 | 0.19 | 0.94 | 0.67 | 0.62 | 0.75 | 0.10 | 0.69 | 0.69 | 0.91 | 0.77 | 0.04 | 0.70 | 0.90 | 0.98 | 0.86 |
| ACUTE pca20 | 0.19 | 0.94 | 0.69 | 0.62 | 0.75 | 0.11 | 0.69 | 0.69 | 0.91 | 0.76 | 0.04 | 0.70 | 0.90 | 0.98 | 0.86 |
| ACUTE early act | 0.19 | 0.94 | 0.70 | 0.62 | 0.75 | 0.10 | 0.68 | 0.69 | 0.91 | 0.76 | 0.04 | 0.71 | 0.90 | 0.98 | 0.86 |
| ACUTE mid act | 0.19 | 0.94 | 0.70 | 0.62 | 0.76 | 0.10 | 0.70 | 0.69 | 0.91 | 0.77 | 0.04 | 0.72 | 0.90 | 0.98 | 0.86 |
| ACUTE late act | 0.19 | 0.94 | 0.70 | 0.62 | 0.76 | 0.10 | 0.68 | 0.69 | 0.91 | 0.76 | 0.03 | 0.71 | 0.90 | 0.98 | 0.86 |

*Table 8.* Results on the SCITLDR dev set evaluated at different Rouge-L thresholds to test threshold sensitivity on Qwen3-14B. Lower smECE is better, while higher AUC-EURO is better.

| | | Threshold = 0.2 | | | | | Threshold = 0.3 | | | | | Threshold = 0.4 | | | |
|---|---|---|---|---|---|---|---|---|---|---|---|---|---|---|---|
| | | AUC-EURO (↑) | | | | | AUC-EURO (↑) | | | | | AUC-EURO (↑) | | | |
| estimator | smECE (↓) | low | med | high | all | smECE (↓) | low | med | high | all | smECE (↓) | low | med | high | all |
| Raw Conf | 0.33 | 0.08 | 0.24 | 0.62 | 0.31 | 0.14 | 0.32 | 0.70 | 0.92 | 0.64 | 0.03 | 0.71 | 0.95 | 0.99 | 0.88 |
| HRE | 0.19 | 0.93 | 0.66 | 0.62 | 0.74 | 0.09 | 0.66 | 0.70 | 0.92 | 0.76 | 0.00 | 0.78 | 0.95 | 0.99 | 0.90 |
| NWKR | 0.19 | 0.94 | 0.67 | 0.62 | 0.74 | 0.09 | 0.67 | 0.70 | 0.92 | 0.76 | 0.01 | 0.78 | 0.95 | 0.99 | 0.90 |
| Platt | 0.19 | 0.94 | 0.67 | 0.62 | 0.75 | 0.09 | 0.67 | 0.70 | 0.92 | 0.76 | 0.01 | 0.78 | 0.95 | 0.99 | 0.90 |
| Isotonic | 0.18 | 0.94 | 0.67 | 0.61 | 0.74 | 0.09 | 0.67 | 0.70 | 0.92 | 0.76 | 0.01 | 0.77 | 0.95 | 0.99 | 0.90 |
| Beta | 0.19 | 0.94 | 0.67 | 0.62 | 0.75 | 0.09 | 0.67 | 0.70 | 0.92 | 0.76 | 0.01 | 0.78 | 0.95 | 0.99 | 0.90 |
| ACUTE cosine | 0.19 | 0.94 | 0.64 | 0.62 | 0.74 | 0.09 | 0.67 | 0.70 | 0.92 | 0.76 | 0.02 | 0.77 | 0.95 | 0.99 | 0.90 |
| ACUTE pca10 | 0.19 | 0.94 | 0.65 | 0.63 | 0.74 | 0.10 | 0.69 | 0.70 | 0.92 | 0.77 | 0.02 | 0.77 | 0.95 | 0.99 | 0.90 |
| ACUTE pca20 | 0.19 | 0.94 | 0.67 | 0.63 | 0.75 | 0.10 | 0.68 | 0.71 | 0.92 | 0.77 | 0.02 | 0.77 | 0.95 | 0.99 | 0.90 |
| ACUTE early act | 0.18 | 0.94 | 0.69 | 0.63 | 0.75 | 0.09 | 0.69 | 0.70 | 0.92 | 0.77 | 0.01 | 0.78 | 0.95 | 0.99 | 0.90 |
| ACUTE mid act | 0.18 | 0.94 | 0.69 | 0.63 | 0.75 | 0.09 | 0.69 | 0.71 | 0.92 | 0.77 | 0.01 | 0.78 | 0.95 | 0.99 | 0.90 |
| ACUTE late act | 0.18 | 0.94 | 0.68 | 0.63 | 0.75 | 0.09 | 0.68 | 0.70 | 0.92 | 0.77 | 0.01 | 0.77 | 0.95 | 0.99 | 0.90 |

*Table 9.* Results on the SCITLDR dev set evaluated at different Rouge-L thresholds to test threshold sensitivity on phi-4. Lower smECE is better, while higher AUC-EURO is better.

| estimator | smECE (↓) | Threshold = 0.2 AUC-EURO (↑) low | med | high | all | smECE (↓) | Threshold = 0.3 AUC-EURO (↑) low | med | high | all | smECE (↓) | Threshold = 0.4 AUC-EURO (↑) low | med | high | all |
|---|---|---|---|---|---|---|---|---|---|---|---|---|---|---|---|
| Raw Conf | 0.28 | 0.12 | 0.39 | 0.76 | 0.42 | 0.10 | 0.42 | 0.80 | 0.95 | 0.72 | 0.03 | 0.69 | 0.94 | 0.99 | 0.87 |
| HRE | 0.15 | 0.89 | 0.55 | 0.76 | 0.73 | 0.05 | 0.66 | 0.80 | 0.95 | 0.80 | 0.01 | 0.76 | 0.94 | 0.99 | 0.90 |
| NWKR | 0.15 | 0.89 | 0.55 | 0.76 | 0.73 | 0.04 | 0.66 | 0.80 | 0.95 | 0.80 | 0.01 | 0.76 | 0.94 | 0.99 | 0.90 |
| Platt | 0.15 | 0.89 | 0.55 | 0.76 | 0.73 | 0.04 | 0.66 | 0.80 | 0.95 | 0.80 | 0.01 | 0.76 | 0.94 | 0.99 | 0.90 |
| Isotonic | 0.15 | 0.89 | 0.55 | 0.76 | 0.73 | 0.05 | 0.66 | 0.80 | 0.95 | 0.80 | 0.01 | 0.76 | 0.94 | 0.99 | 0.90 |
| Beta | 0.15 | 0.89 | 0.55 | 0.76 | 0.73 | 0.04 | 0.66 | 0.80 | 0.95 | 0.80 | 0.01 | 0.76 | 0.94 | 0.99 | 0.90 |
| ACUTE cosine | 0.15 | 0.88 | 0.59 | 0.76 | 0.74 | 0.07 | 0.68 | 0.79 | 0.95 | 0.80 | 0.03 | 0.76 | 0.94 | 0.99 | 0.89 |
| ACUTE pca10 | 0.16 | 0.88 | 0.60 | 0.76 | 0.75 | 0.06 | 0.70 | 0.80 | 0.95 | 0.82 | 0.03 | 0.76 | 0.94 | 0.99 | 0.89 |
| ACUTE pca20 | 0.15 | 0.89 | 0.61 | 0.76 | 0.76 | 0.06 | 0.70 | 0.80 | 0.95 | 0.82 | 0.03 | 0.76 | 0.94 | 0.99 | 0.89 |
| ACUTE early act | 0.16 | 0.89 | 0.60 | 0.76 | 0.75 | 0.05 | 0.70 | 0.80 | 0.95 | 0.82 | 0.02 | 0.77 | 0.94 | 0.99 | 0.90 |
| ACUTE mid act | 0.15 | 0.89 | 0.62 | 0.76 | 0.76 | 0.05 | 0.70 | 0.80 | 0.95 | 0.82 | 0.02 | 0.78 | 0.94 | 0.99 | 0.90 |
| ACUTE late act | 0.15 | 0.89 | 0.60 | 0.76 | 0.75 | 0.05 | 0.69 | 0.80 | 0.95 | 0.81 | 0.02 | 0.77 | 0.94 | 0.99 | 0.90 |

*Table 10.* Results on the SCITLDR dev set evaluated at different Rouge-L thresholds to test threshold sensitivity on SmolLM3-3B. Lower smECE is better, while higher AUC-EURO is better.

| estimator | smECE (↓) | Threshold = 0.2 AUC-EURO (↑) low | med | high | all | smECE (↓) | Threshold = 0.3 AUC-EURO (↑) low | med | high | all | smECE (↓) | Threshold = 0.4 AUC-EURO (↑) low | med | high | all |
|---|---|---|---|---|---|---|---|---|---|---|---|---|---|---|---|
| Raw Conf | 0.39 | 0.14 | 0.30 | 0.68 | 0.37 | 0.15 | 0.41 | 0.76 | 0.94 | 0.70 | 0.02 | 0.76 | 0.96 | 0.99 | 0.90 |
| HRE | 0.20 | 0.92 | 0.58 | 0.69 | 0.73 | 0.09 | 0.64 | 0.76 | 0.94 | 0.78 | 0.01 | 0.81 | 0.96 | 0.99 | 0.92 |
| NWKR | 0.19 | 0.92 | 0.59 | 0.69 | 0.73 | 0.08 | 0.64 | 0.76 | 0.94 | 0.78 | 0.01 | 0.81 | 0.96 | 0.99 | 0.92 |
| Platt | 0.19 | 0.92 | 0.58 | 0.69 | 0.73 | 0.08 | 0.64 | 0.76 | 0.94 | 0.78 | 0.00 | 0.81 | 0.96 | 0.99 | 0.92 |
| Isotonic | 0.19 | 0.92 | 0.59 | 0.69 | 0.73 | 0.08 | 0.64 | 0.76 | 0.94 | 0.78 | 0.01 | 0.81 | 0.96 | 0.99 | 0.92 |
| Beta | 0.19 | 0.92 | 0.59 | 0.68 | 0.73 | 0.08 | 0.64 | 0.76 | 0.94 | 0.78 | 0.01 | 0.81 | 0.96 | 0.99 | 0.92 |
| ACUTE cosine | 0.20 | 0.92 | 0.58 | 0.69 | 0.73 | 0.09 | 0.63 | 0.76 | 0.94 | 0.78 | 0.03 | 0.77 | 0.96 | 0.99 | 0.91 |
| ACUTE pca10 | 0.19 | 0.92 | 0.61 | 0.69 | 0.74 | 0.09 | 0.64 | 0.76 | 0.94 | 0.78 | 0.02 | 0.80 | 0.96 | 0.99 | 0.92 |
| ACUTE pca20 | 0.19 | 0.92 | 0.61 | 0.69 | 0.74 | 0.09 | 0.65 | 0.76 | 0.94 | 0.78 | 0.02 | 0.80 | 0.96 | 0.99 | 0.92 |
| ACUTE early act | 0.19 | 0.92 | 0.61 | 0.69 | 0.74 | 0.09 | 0.65 | 0.76 | 0.94 | 0.78 | 0.01 | 0.82 | 0.96 | 0.99 | 0.92 |
| ACUTE mid act | 0.19 | 0.92 | 0.62 | 0.69 | 0.74 | 0.09 | 0.65 | 0.76 | 0.94 | 0.78 | 0.01 | 0.82 | 0.96 | 0.99 | 0.92 |
| ACUTE late act | 0.19 | 0.92 | 0.61 | 0.69 | 0.74 | 0.09 | 0.65 | 0.76 | 0.94 | 0.78 | 0.01 | 0.81 | 0.96 | 0.99 | 0.92 |

*Table 11.* Results on the SCITLDR dev set evaluated at different Rouge-L thresholds to test threshold sensitivity on Qwen3-30B-A3B-Instruct-2507. Lower smECE is better, while higher AUC-EURO is better.

| estimator | APIGen smECE ($\downarrow$) | AUC-EURO ($\uparrow$) low | med | high | all | SCITLDR smECE ($\downarrow$) | AUC-EURO ($\uparrow$) low | med | high | all |
|---|---|---|---|---|---|---|---|---|---|---|
| Raw Conf | 0.213 | 0.246 | 0.528 | 0.827 | 0.531 | 0.191 | 0.324 | 0.671 | 0.908 | 0.631 |
| HRE | 0.019 | 0.846 | 0.612 | 0.832 | 0.764 | 0.095 | 0.690 | 0.671 | 0.908 | 0.756 |
| NWKR | 0.039 | 0.831 | 0.595 | 0.834 | 0.754 | 0.100 | 0.687 | 0.671 | 0.908 | 0.754 |
| Platt | 0.037 | 0.826 | 0.607 | 0.834 | 0.756 | 0.096 | 0.685 | 0.671 | 0.908 | 0.754 |
| Isotonic | 0.014 | 0.831 | 0.590 | 0.834 | 0.752 | 0.096 | 0.691 | 0.671 | 0.908 | 0.756 |
| Beta | 0.021 | 0.831 | 0.587 | 0.834 | 0.751 | 0.096 | 0.686 | 0.671 | 0.908 | 0.754 |
| ACUTE cosine | 0.052 | 0.859 | 0.705 | 0.833 | 0.800 | 0.100 | 0.688 | 0.668 | 0.908 | 0.754 |
| ACUTE cosine Platt | 0.046 | 0.859 | 0.702 | 0.839 | 0.801 | 0.111 | 0.682 | 0.670 | 0.908 | 0.753 |
| ACUTE cosine Isotonic | 0.046 | 0.861 | 0.703 | 0.838 | 0.801 | 0.116 | 0.677 | 0.666 | 0.908 | 0.749 |
| ACUTE cosine Beta | 0.032 | 0.861 | 0.706 | 0.839 | 0.803 | 0.111 | 0.682 | 0.671 | 0.908 | 0.753 |
| ACUTE pca10 | 0.032 | 0.912 | 0.820 | 0.879 | 0.871 | 0.108 | 0.698 | 0.676 | 0.908 | 0.760 |
| ACUTE pca10 Platt | 0.024 | 0.916 | 0.819 | 0.879 | 0.872 | 0.107 | 0.700 | 0.675 | 0.908 | 0.760 |
| ACUTE pca10 Isotonic | 0.026 | 0.913 | 0.819 | 0.877 | 0.870 | 0.107 | 0.687 | 0.676 | 0.908 | 0.757 |
| ACUTE pca10 Beta | 0.024 | 0.917 | 0.819 | 0.879 | 0.872 | 0.107 | 0.700 | 0.674 | 0.908 | 0.760 |
| ACUTE pca20 | 0.048 | 0.912 | 0.825 | 0.871 | 0.870 | 0.113 | 0.702 | 0.674 | 0.908 | 0.761 |
| ACUTE pca20 Platt | 0.036 | 0.919 | 0.823 | 0.872 | 0.872 | 0.109 | 0.705 | 0.675 | 0.908 | 0.762 |
| ACUTE pca20 Isotonic | 0.025 | 0.921 | 0.815 | 0.867 | 0.868 | 0.111 | 0.697 | 0.674 | 0.909 | 0.759 |
| ACUTE pca20 Beta | 0.027 | 0.921 | 0.824 | 0.874 | 0.873 | 0.110 | 0.706 | 0.675 | 0.908 | 0.762 |
| ACUTE early act | 0.078 | 0.894 | 0.805 | 0.865 | 0.854 | 0.105 | 0.699 | 0.672 | 0.908 | 0.759 |
| ACUTE early act Platt | 0.036 | 0.909 | 0.806 | 0.872 | 0.863 | 0.112 | 0.698 | 0.672 | 0.908 | 0.759 |
| ACUTE early act Isotonic | 0.020 | 0.908 | 0.795 | 0.866 | 0.857 | 0.106 | 0.691 | 0.672 | 0.908 | 0.756 |
| ACUTE early act Beta | 0.032 | 0.911 | 0.806 | 0.872 | 0.863 | 0.112 | 0.698 | 0.672 | 0.908 | 0.759 |
| ACUTE mid act | 0.086 | 0.895 | 0.826 | 0.873 | 0.864 | 0.103 | 0.705 | 0.672 | 0.908 | 0.761 |
| ACUTE mid act Platt | 0.031 | 0.913 | 0.832 | 0.885 | 0.877 | 0.113 | 0.701 | 0.677 | 0.908 | 0.761 |
| ACUTE mid act Isotonic | 0.046 | 0.912 | 0.822 | 0.877 | 0.871 | 0.113 | 0.685 | 0.679 | 0.908 | 0.756 |
| ACUTE mid act Beta | 0.030 | 0.915 | 0.832 | 0.885 | 0.878 | 0.114 | 0.701 | 0.677 | 0.908 | 0.761 |
| ACUTE late act | 0.084 | 0.902 | 0.831 | 0.874 | 0.869 | 0.104 | 0.692 | 0.671 | 0.908 | 0.756 |
| ACUTE late act Platt | 0.035 | 0.919 | 0.836 | 0.882 | 0.880 | 0.110 | 0.686 | 0.671 | 0.908 | 0.754 |
| ACUTE late act Isotonic | 0.051 | 0.908 | 0.824 | 0.877 | 0.870 | 0.113 | 0.681 | 0.671 | 0.908 | 0.753 |
| ACUTE late act Beta | 0.026 | 0.920 | 0.836 | 0.884 | 0.880 | 0.110 | 0.686 | 0.671 | 0.908 | 0.754 |

*Table 12*. Results on the APIGen test subset (left), and on the SCITLDR dev set (right) on gemma3-12b-it. Lower smECE is better, while higher AUC-EURO is better.

| | APIGen | | | | | SCITLDR | | | | |
| | | AUC-EURO ($\uparrow$) | | | | | AUC-EURO ($\uparrow$) | | | |
| estimator | smECE ($\downarrow$) | low | med | high | all | smECE ($\downarrow$) | low | med | high | all |
|---|---|---|---|---|---|---|---|---|---|---|
| Raw Conf | 0.206 | 0.260 | 0.550 | 0.855 | 0.552 | 0.165 | 0.356 | 0.710 | 0.922 | 0.660 |
| HRE | 0.032 | 0.771 | 0.624 | 0.878 | 0.758 | 0.080 | 0.666 | 0.710 | 0.922 | 0.765 |
| NWKR | 0.039 | 0.776 | 0.619 | 0.878 | 0.758 | 0.075 | 0.672 | 0.710 | 0.922 | 0.767 |
| Platt | 0.031 | 0.779 | 0.627 | 0.878 | 0.761 | 0.074 | 0.673 | 0.710 | 0.922 | 0.768 |
| Isotonic | 0.032 | 0.776 | 0.608 | 0.878 | 0.754 | 0.076 | 0.676 | 0.710 | 0.922 | 0.768 |
| Beta | 0.042 | 0.767 | 0.620 | 0.878 | 0.755 | 0.074 | 0.674 | 0.710 | 0.922 | 0.768 |
| ACUTE cosine | 0.034 | 0.791 | 0.678 | 0.879 | 0.783 | 0.082 | 0.654 | 0.708 | 0.922 | 0.760 |
| ACUTE cosine Platt | 0.038 | 0.793 | 0.676 | 0.878 | 0.783 | 0.036 | 0.687 | 0.710 | 0.922 | 0.772 |
| ACUTE cosine Isotonic | 0.058 | 0.791 | 0.667 | 0.871 | 0.777 | 0.052 | 0.680 | 0.706 | 0.915 | 0.766 |
| ACUTE cosine Beta | 0.040 | 0.793 | 0.676 | 0.878 | 0.782 | 0.041 | 0.686 | 0.709 | 0.922 | 0.772 |
| ACUTE pca10 | 0.043 | 0.825 | 0.726 | 0.883 | 0.812 | 0.088 | 0.683 | 0.715 | 0.922 | 0.773 |
| ACUTE pca10 Platt | 0.033 | 0.826 | 0.732 | 0.882 | 0.813 | 0.041 | 0.707 | 0.713 | 0.922 | 0.780 |
| ACUTE pca10 Isotonic | 0.045 | 0.816 | 0.727 | 0.878 | 0.807 | 0.039 | 0.701 | 0.713 | 0.922 | 0.778 |
| ACUTE pca10 Beta | 0.033 | 0.826 | 0.732 | 0.881 | 0.813 | 0.041 | 0.707 | 0.713 | 0.922 | 0.780 |
| ACUTE pca20 | 0.088 | 0.850 | 0.772 | 0.886 | 0.836 | 0.091 | 0.690 | 0.714 | 0.922 | 0.774 |
| ACUTE pca20 Platt | 0.033 | 0.860 | 0.787 | 0.894 | 0.847 | 0.044 | 0.713 | 0.715 | 0.922 | 0.783 |
| ACUTE pca20 Isotonic | 0.036 | 0.851 | 0.787 | 0.893 | 0.844 | 0.043 | 0.710 | 0.717 | 0.922 | 0.782 |
| ACUTE pca20 Beta | 0.032 | 0.860 | 0.787 | 0.895 | 0.848 | 0.043 | 0.714 | 0.714 | 0.922 | 0.783 |
| ACUTE early act | 0.037 | 0.834 | 0.724 | 0.885 | 0.812 | 0.083 | 0.696 | 0.712 | 0.922 | 0.775 |
| ACUTE early act Platt | 0.035 | 0.841 | 0.728 | 0.882 | 0.817 | 0.048 | 0.712 | 0.712 | 0.922 | 0.781 |
| ACUTE early act Isotonic | 0.028 | 0.840 | 0.726 | 0.879 | 0.815 | 0.046 | 0.718 | 0.714 | 0.922 | 0.784 |
| ACUTE early act Beta | 0.034 | 0.841 | 0.728 | 0.883 | 0.817 | 0.048 | 0.713 | 0.712 | 0.922 | 0.782 |
| ACUTE mid act | 0.063 | 0.840 | 0.750 | 0.883 | 0.824 | 0.081 | 0.690 | 0.713 | 0.922 | 0.774 |
| ACUTE mid act Platt | 0.035 | 0.851 | 0.758 | 0.890 | 0.833 | 0.050 | 0.708 | 0.717 | 0.922 | 0.781 |
| ACUTE mid act Isotonic | 0.026 | 0.846 | 0.760 | 0.886 | 0.831 | 0.052 | 0.707 | 0.713 | 0.922 | 0.780 |
| ACUTE mid act Beta | 0.033 | 0.851 | 0.759 | 0.891 | 0.834 | 0.050 | 0.708 | 0.716 | 0.922 | 0.781 |
| ACUTE late act | 0.076 | 0.843 | 0.752 | 0.883 | 0.826 | 0.081 | 0.688 | 0.712 | 0.922 | 0.773 |
| ACUTE late act Platt | 0.033 | 0.855 | 0.763 | 0.889 | 0.836 | 0.042 | 0.705 | 0.714 | 0.922 | 0.780 |
| ACUTE late act Isotonic | 0.034 | 0.850 | 0.759 | 0.889 | 0.833 | 0.043 | 0.705 | 0.710 | 0.922 | 0.778 |
| ACUTE late act Beta | 0.027 | 0.855 | 0.765 | 0.890 | 0.837 | 0.042 | 0.706 | 0.713 | 0.922 | 0.780 |

*Table 13.* Results on the APIGen test subset (left), and on the SCITLDR dev set (right) on gemma3-4b-it. Lower smECE is better, while higher AUC-EURO is better.

| | APIGen | | | | | SCITLDR | | | | |
| | | AUC-EURO (↑) | | | | | AUC-EURO (↑) | | | |
| estimator | smECE (↓) | low | med | high | all | smECE (↓) | low | med | high | all |
|---|---|---|---|---|---|---|---|---|---|---|
| Raw Conf | 0.277 | 0.226 | 0.354 | 0.664 | 0.413 | 0.159 | 0.385 | 0.712 | 0.923 | 0.670 |
| HRE | 0.019 | 0.938 | 0.760 | 0.685 | 0.796 | 0.074 | 0.665 | 0.715 | 0.923 | 0.767 |
| NWKR | 0.024 | 0.933 | 0.737 | 0.674 | 0.783 | 0.076 | 0.673 | 0.714 | 0.923 | 0.769 |
| Platt | 0.024 | 0.933 | 0.733 | 0.683 | 0.784 | 0.073 | 0.670 | 0.714 | 0.923 | 0.768 |
| Isotonic | 0.010 | 0.933 | 0.736 | 0.674 | 0.783 | 0.073 | 0.676 | 0.714 | 0.923 | 0.770 |
| Beta | 0.016 | 0.933 | 0.736 | 0.669 | 0.781 | 0.073 | 0.672 | 0.714 | 0.923 | 0.769 |
| ACUTE cosine | 0.030 | 0.931 | 0.755 | 0.729 | 0.806 | 0.080 | 0.681 | 0.706 | 0.923 | 0.769 |
| ACUTE cosine Platt | 0.031 | 0.932 | 0.755 | 0.728 | 0.806 | 0.075 | 0.680 | 0.714 | 0.923 | 0.771 |
| ACUTE cosine Isotonic | 0.036 | 0.926 | 0.753 | 0.723 | 0.802 | 0.072 | 0.680 | 0.713 | 0.923 | 0.771 |
| ACUTE cosine Beta | 0.031 | 0.932 | 0.755 | 0.728 | 0.806 | 0.075 | 0.680 | 0.713 | 0.923 | 0.771 |
| ACUTE pca10 | 0.032 | 0.939 | 0.807 | 0.792 | 0.847 | 0.081 | 0.695 | 0.713 | 0.923 | 0.776 |
| ACUTE pca10 Platt | 0.036 | 0.940 | 0.806 | 0.792 | 0.847 | 0.082 | 0.691 | 0.713 | 0.923 | 0.775 |
| ACUTE pca10 Isotonic | 0.047 | 0.931 | 0.805 | 0.791 | 0.843 | 0.087 | 0.688 | 0.712 | 0.923 | 0.773 |
| ACUTE pca10 Beta | 0.037 | 0.939 | 0.805 | 0.793 | 0.847 | 0.082 | 0.691 | 0.713 | 0.923 | 0.775 |
| ACUTE pca20 | 0.040 | 0.938 | 0.812 | 0.795 | 0.849 | 0.080 | 0.698 | 0.717 | 0.923 | 0.779 |
| ACUTE pca20 Platt | 0.031 | 0.941 | 0.815 | 0.795 | 0.851 | 0.081 | 0.698 | 0.717 | 0.923 | 0.779 |
| ACUTE pca20 Isotonic | 0.049 | 0.936 | 0.814 | 0.791 | 0.848 | 0.079 | 0.700 | 0.719 | 0.923 | 0.780 |
| ACUTE pca20 Beta | 0.030 | 0.940 | 0.814 | 0.796 | 0.851 | 0.080 | 0.699 | 0.716 | 0.923 | 0.779 |
| ACUTE early act | 0.028 | 0.935 | 0.790 | 0.775 | 0.834 | 0.078 | 0.687 | 0.713 | 0.923 | 0.773 |
| ACUTE early act Platt | 0.029 | 0.937 | 0.793 | 0.779 | 0.837 | 0.082 | 0.686 | 0.713 | 0.923 | 0.773 |
| ACUTE early act Isotonic | 0.034 | 0.936 | 0.790 | 0.776 | 0.835 | 0.080 | 0.680 | 0.714 | 0.923 | 0.771 |
| ACUTE early act Beta | 0.030 | 0.937 | 0.793 | 0.778 | 0.837 | 0.081 | 0.687 | 0.713 | 0.923 | 0.773 |
| ACUTE mid act | 0.048 | 0.937 | 0.824 | 0.818 | 0.860 | 0.076 | 0.688 | 0.714 | 0.923 | 0.774 |
| ACUTE mid act Platt | 0.024 | 0.943 | 0.834 | 0.832 | 0.871 | 0.080 | 0.685 | 0.713 | 0.923 | 0.773 |
| ACUTE mid act Isotonic | 0.032 | 0.942 | 0.833 | 0.827 | 0.868 | 0.083 | 0.678 | 0.713 | 0.923 | 0.770 |
| ACUTE mid act Beta | 0.024 | 0.943 | 0.834 | 0.832 | 0.871 | 0.079 | 0.685 | 0.714 | 0.923 | 0.773 |
| ACUTE late act | 0.055 | 0.937 | 0.821 | 0.811 | 0.857 | 0.076 | 0.681 | 0.714 | 0.923 | 0.772 |
| ACUTE late act Platt | 0.028 | 0.943 | 0.831 | 0.821 | 0.866 | 0.083 | 0.669 | 0.716 | 0.923 | 0.768 |
| ACUTE late act Isotonic | 0.023 | 0.943 | 0.832 | 0.821 | 0.866 | 0.085 | 0.661 | 0.717 | 0.923 | 0.766 |
| ACUTE late act Beta | 0.029 | 0.943 | 0.831 | 0.821 | 0.866 | 0.083 | 0.669 | 0.716 | 0.923 | 0.768 |

*Table 14.* Results on the APIGen test subset (left), and on the SCITLDR dev set (right) on Qwen3-4B-Instruct-2507. Lower smECE is better, while higher AUC-EURO is better.

| | APIGen | | | | | SCITLDR | | | | |
|---|---|---|---|---|---|---|---|---|---|---|
| | | AUC-EURO (↑) | | | | | AUC-EURO (↑) | | | |
| estimator | smECE (↓) | low | med | high | all | smECE (↓) | low | med | high | all |
| Raw Conf | 0.315 | 0.196 | 0.308 | 0.593 | 0.364 | 0.164 | 0.344 | 0.689 | 0.914 | 0.646 |
| HRE | 0.036 | 0.922 | 0.769 | 0.690 | 0.795 | 0.094 | 0.675 | 0.688 | 0.914 | 0.758 |
| NWKR | 0.038 | 0.932 | 0.739 | 0.685 | 0.787 | 0.105 | 0.672 | 0.688 | 0.914 | 0.757 |
| Platt | 0.043 | 0.932 | 0.731 | 0.691 | 0.786 | 0.094 | 0.682 | 0.688 | 0.914 | 0.760 |
| Isotonic | 0.028 | 0.932 | 0.733 | 0.680 | 0.783 | 0.095 | 0.673 | 0.688 | 0.914 | 0.758 |
| Beta | 0.025 | 0.932 | 0.733 | 0.673 | 0.781 | 0.094 | 0.679 | 0.688 | 0.914 | 0.760 |
| ACUTE cosine | 0.036 | 0.934 | 0.770 | 0.739 | 0.816 | 0.094 | 0.675 | 0.687 | 0.914 | 0.758 |
| ACUTE cosine Platt | 0.023 | 0.934 | 0.773 | 0.741 | 0.817 | 0.089 | 0.684 | 0.688 | 0.914 | 0.761 |
| ACUTE cosine Isotonic | 0.026 | 0.932 | 0.763 | 0.737 | 0.812 | 0.094 | 0.679 | 0.685 | 0.914 | 0.758 |
| ACUTE cosine Beta | 0.022 | 0.934 | 0.773 | 0.742 | 0.817 | 0.089 | 0.684 | 0.688 | 0.914 | 0.761 |
| ACUTE pca10 | 0.039 | 0.937 | 0.816 | 0.800 | 0.852 | 0.102 | 0.694 | 0.692 | 0.914 | 0.766 |
| ACUTE pca10 Platt | 0.028 | 0.938 | 0.817 | 0.802 | 0.853 | 0.095 | 0.698 | 0.689 | 0.914 | 0.766 |
| ACUTE pca10 Isotonic | 0.052 | 0.930 | 0.812 | 0.792 | 0.845 | 0.095 | 0.685 | 0.696 | 0.914 | 0.764 |
| ACUTE pca10 Beta | 0.029 | 0.938 | 0.817 | 0.800 | 0.853 | 0.095 | 0.697 | 0.688 | 0.914 | 0.766 |
| ACUTE pca20 | 0.052 | 0.938 | 0.823 | 0.810 | 0.858 | 0.106 | 0.687 | 0.690 | 0.914 | 0.763 |
| ACUTE pca20 Platt | 0.027 | 0.941 | 0.823 | 0.820 | 0.862 | 0.093 | 0.697 | 0.688 | 0.914 | 0.766 |
| ACUTE pca20 Isotonic | 0.041 | 0.936 | 0.823 | 0.807 | 0.856 | 0.092 | 0.687 | 0.693 | 0.914 | 0.764 |
| ACUTE pca20 Beta | 0.031 | 0.941 | 0.823 | 0.817 | 0.861 | 0.093 | 0.696 | 0.688 | 0.914 | 0.765 |
| ACUTE early act | 0.053 | 0.934 | 0.796 | 0.776 | 0.836 | 0.102 | 0.681 | 0.690 | 0.914 | 0.760 |
| ACUTE early act Platt | 0.025 | 0.937 | 0.800 | 0.791 | 0.844 | 0.093 | 0.685 | 0.688 | 0.914 | 0.762 |
| ACUTE early act Isotonic | 0.035 | 0.936 | 0.799 | 0.784 | 0.840 | 0.096 | 0.683 | 0.691 | 0.914 | 0.762 |
| ACUTE early act Beta | 0.024 | 0.937 | 0.800 | 0.791 | 0.844 | 0.094 | 0.685 | 0.689 | 0.914 | 0.762 |
| ACUTE mid act | 0.064 | 0.937 | 0.834 | 0.815 | 0.863 | 0.098 | 0.696 | 0.689 | 0.914 | 0.766 |
| ACUTE mid act Platt | 0.022 | 0.945 | 0.842 | 0.835 | 0.875 | 0.099 | 0.697 | 0.691 | 0.914 | 0.767 |
| ACUTE mid act Isotonic | 0.023 | 0.943 | 0.839 | 0.836 | 0.873 | 0.103 | 0.692 | 0.693 | 0.914 | 0.765 |
| ACUTE mid act Beta | 0.023 | 0.944 | 0.842 | 0.835 | 0.874 | 0.099 | 0.698 | 0.691 | 0.914 | 0.767 |
| ACUTE late act | 0.078 | 0.936 | 0.831 | 0.817 | 0.862 | 0.099 | 0.681 | 0.689 | 0.914 | 0.761 |
| ACUTE late act Platt | 0.028 | 0.943 | 0.839 | 0.838 | 0.874 | 0.096 | 0.680 | 0.690 | 0.914 | 0.761 |
| ACUTE late act Isotonic | 0.032 | 0.942 | 0.835 | 0.836 | 0.872 | 0.092 | 0.662 | 0.690 | 0.914 | 0.754 |
| ACUTE late act Beta | 0.031 | 0.942 | 0.839 | 0.836 | 0.873 | 0.095 | 0.679 | 0.690 | 0.914 | 0.760 |

*Table 15*. Results on the APIGen test subset (left), and on the SCITLDR dev set (right) on Qwen3-14B. Lower smECE is better, while higher AUC-EURO is better.

| | | APIGen | | | | | SCITLDR | | | |
| | | AUC-EURO (↑) | | | | | AUC-EURO (↑) | | | |
| estimator | smECE (↓) | low | med | high | all | smECE (↓) | low | med | high | all |
|---|---|---|---|---|---|---|---|---|---|---|
| Raw Conf | 0.210 | 0.216 | 0.566 | 0.864 | 0.545 | 0.143 | 0.323 | 0.703 | 0.920 | 0.645 |
| HRE | 0.020 | 0.789 | 0.595 | 0.862 | 0.749 | 0.090 | 0.665 | 0.702 | 0.918 | 0.761 |
| NWKR | 0.014 | 0.796 | 0.595 | 0.864 | 0.752 | 0.087 | 0.672 | 0.703 | 0.920 | 0.764 |
| Platt | 0.049 | 0.827 | 0.666 | 0.864 | 0.786 | 0.088 | 0.670 | 0.704 | 0.920 | 0.764 |
| Isotonic | 0.008 | 0.796 | 0.594 | 0.860 | 0.751 | 0.087 | 0.674 | 0.704 | 0.920 | 0.765 |
| Beta | 0.009 | 0.796 | 0.596 | 0.864 | 0.753 | 0.087 | 0.672 | 0.704 | 0.920 | 0.764 |
| ACUTE cosine | 0.042 | 0.912 | 0.817 | 0.884 | 0.872 | 0.087 | 0.672 | 0.702 | 0.920 | 0.764 |
| ACUTE cosine Platt | 0.038 | 0.912 | 0.821 | 0.883 | 0.872 | 0.094 | 0.671 | 0.703 | 0.920 | 0.764 |
| ACUTE cosine Isotonic | 0.073 | 0.907 | 0.820 | 0.863 | 0.864 | 0.111 | 0.655 | 0.694 | 0.920 | 0.755 |
| ACUTE cosine Beta | 0.057 | 0.911 | 0.820 | 0.875 | 0.870 | 0.096 | 0.667 | 0.702 | 0.920 | 0.762 |
| ACUTE pca10 | 0.028 | 0.936 | 0.858 | 0.905 | 0.900 | 0.097 | 0.686 | 0.704 | 0.920 | 0.769 |
| ACUTE pca10 Platt | 0.031 | 0.937 | 0.860 | 0.904 | 0.901 | 0.095 | 0.688 | 0.705 | 0.920 | 0.770 |
| ACUTE pca10 Isotonic | 0.031 | 0.935 | 0.860 | 0.902 | 0.899 | 0.089 | 0.674 | 0.704 | 0.920 | 0.765 |
| ACUTE pca10 Beta | 0.030 | 0.937 | 0.860 | 0.904 | 0.901 | 0.095 | 0.686 | 0.705 | 0.920 | 0.770 |
| ACUTE pca20 | 0.052 | 0.939 | 0.890 | 0.919 | 0.916 | 0.098 | 0.684 | 0.706 | 0.920 | 0.769 |
| ACUTE pca20 Platt | 0.024 | 0.948 | 0.891 | 0.921 | 0.921 | 0.090 | 0.688 | 0.706 | 0.920 | 0.771 |
| ACUTE pca20 Isotonic | 0.026 | 0.939 | 0.891 | 0.922 | 0.918 | 0.086 | 0.676 | 0.704 | 0.920 | 0.766 |
| ACUTE pca20 Beta | 0.022 | 0.948 | 0.893 | 0.921 | 0.921 | 0.090 | 0.689 | 0.705 | 0.920 | 0.771 |
| ACUTE early act | 0.052 | 0.934 | 0.877 | 0.908 | 0.904 | 0.092 | 0.686 | 0.705 | 0.920 | 0.770 |
| ACUTE early act Platt | 0.041 | 0.943 | 0.882 | 0.908 | 0.911 | 0.090 | 0.685 | 0.709 | 0.920 | 0.770 |
| ACUTE early act Isotonic | 0.033 | 0.946 | 0.880 | 0.908 | 0.912 | 0.093 | 0.681 | 0.706 | 0.919 | 0.768 |
| ACUTE early act Beta | 0.022 | 0.947 | 0.884 | 0.910 | 0.914 | 0.090 | 0.685 | 0.708 | 0.920 | 0.770 |
| ACUTE mid act | 0.048 | 0.936 | 0.875 | 0.916 | 0.909 | 0.091 | 0.688 | 0.706 | 0.920 | 0.770 |
| ACUTE mid act Platt | 0.028 | 0.943 | 0.873 | 0.920 | 0.912 | 0.091 | 0.687 | 0.708 | 0.920 | 0.771 |
| ACUTE mid act Isotonic | 0.052 | 0.936 | 0.867 | 0.908 | 0.904 | 0.095 | 0.682 | 0.704 | 0.920 | 0.768 |
| ACUTE mid act Beta | 0.032 | 0.944 | 0.873 | 0.916 | 0.911 | 0.091 | 0.687 | 0.708 | 0.920 | 0.771 |
| ACUTE late act | 0.042 | 0.936 | 0.875 | 0.916 | 0.909 | 0.091 | 0.680 | 0.705 | 0.920 | 0.767 |
| ACUTE late act Platt | 0.024 | 0.942 | 0.877 | 0.919 | 0.913 | 0.088 | 0.684 | 0.705 | 0.920 | 0.769 |
| ACUTE late act Isotonic | 0.040 | 0.936 | 0.879 | 0.906 | 0.907 | 0.090 | 0.676 | 0.703 | 0.920 | 0.765 |
| ACUTE late act Beta | 0.031 | 0.942 | 0.878 | 0.917 | 0.912 | 0.088 | 0.684 | 0.705 | 0.920 | 0.769 |

*Table 16.* Results on the APIGen test subset (left), and on the SCITLDR dev set (right) on phi-4. Lower smECE is better, while higher AUC-EURO is better.

| | APIGen | | | | | SCITLDR | | | | |
|---|---|---|---|---|---|---|---|---|---|---|
| | | **AUC-EURO** ($\uparrow$) | | | | | **AUC-EURO** ($\uparrow$) | | | |
| **estimator** | **smECE** ($\downarrow$) | low | med | high | all | **smECE** ($\downarrow$) | low | med | high | all |
| Raw Conf | 0.072 | 0.528 | 0.861 | 0.968 | 0.783 | 0.097 | 0.416 | 0.799 | 0.950 | 0.719 |
| HRE | 0.007 | 0.681 | 0.861 | 0.968 | 0.835 | 0.046 | 0.655 | 0.799 | 0.950 | 0.800 |
| NWKR | 0.004 | 0.680 | 0.861 | 0.968 | 0.834 | 0.044 | 0.655 | 0.799 | 0.950 | 0.800 |
| Platt | 0.007 | 0.679 | 0.861 | 0.968 | 0.834 | 0.045 | 0.656 | 0.799 | 0.950 | 0.800 |
| Isotonic | 0.006 | 0.694 | 0.861 | 0.968 | 0.839 | 0.046 | 0.655 | 0.799 | 0.950 | 0.800 |
| Beta | 0.016 | 0.682 | 0.861 | 0.968 | 0.835 | 0.045 | 0.655 | 0.799 | 0.950 | 0.800 |
| ACUTE cosine | 0.023 | 0.757 | 0.864 | 0.968 | 0.862 | 0.066 | 0.676 | 0.791 | 0.950 | 0.804 |
| ACUTE cosine Platt | 0.041 | 0.745 | 0.864 | 0.968 | 0.858 | 0.039 | 0.684 | 0.799 | 0.950 | 0.810 |
| ACUTE cosine Isotonic | 0.035 | 0.750 | 0.863 | 0.968 | 0.859 | 0.039 | 0.690 | 0.799 | 0.950 | 0.812 |
| ACUTE cosine Beta | 0.041 | 0.745 | 0.864 | 0.968 | 0.858 | 0.038 | 0.687 | 0.799 | 0.950 | 0.811 |
| ACUTE pca10 | 0.050 | 0.845 | 0.883 | 0.968 | 0.898 | 0.056 | 0.702 | 0.797 | 0.950 | 0.815 |
| ACUTE pca10 Platt | 0.033 | 0.857 | 0.890 | 0.960 | 0.902 | 0.043 | 0.708 | 0.797 | 0.950 | 0.817 |
| ACUTE pca10 Isotonic | 0.034 | 0.855 | 0.887 | 0.960 | 0.900 | 0.042 | 0.706 | 0.800 | 0.950 | 0.818 |
| ACUTE pca10 Beta | 0.034 | 0.855 | 0.889 | 0.962 | 0.901 | 0.041 | 0.709 | 0.798 | 0.950 | 0.818 |
| ACUTE pca20 | 0.068 | 0.878 | 0.895 | 0.968 | 0.913 | 0.058 | 0.702 | 0.798 | 0.950 | 0.816 |
| ACUTE pca20 Platt | 0.027 | 0.899 | 0.917 | 0.963 | 0.926 | 0.041 | 0.710 | 0.798 | 0.950 | 0.819 |
| ACUTE pca20 Isotonic | 0.041 | 0.890 | 0.899 | 0.963 | 0.917 | 0.037 | 0.707 | 0.799 | 0.950 | 0.818 |
| ACUTE pca20 Beta | 0.028 | 0.898 | 0.916 | 0.964 | 0.926 | 0.039 | 0.712 | 0.798 | 0.950 | 0.819 |
| ACUTE early act | 0.053 | 0.835 | 0.875 | 0.968 | 0.892 | 0.050 | 0.700 | 0.799 | 0.950 | 0.815 |
| ACUTE early act Platt | 0.037 | 0.866 | 0.892 | 0.959 | 0.905 | 0.038 | 0.705 | 0.798 | 0.950 | 0.817 |
| ACUTE early act Isotonic | 0.058 | 0.849 | 0.876 | 0.960 | 0.895 | 0.033 | 0.705 | 0.798 | 0.950 | 0.817 |
| ACUTE early act Beta | 0.039 | 0.866 | 0.890 | 0.960 | 0.905 | 0.036 | 0.707 | 0.798 | 0.950 | 0.818 |
| ACUTE mid act | 0.077 | 0.860 | 0.885 | 0.968 | 0.904 | 0.049 | 0.705 | 0.799 | 0.950 | 0.817 |
| ACUTE mid act Platt | 0.025 | 0.911 | 0.923 | 0.972 | 0.935 | 0.038 | 0.709 | 0.798 | 0.950 | 0.818 |
| ACUTE mid act Isotonic | 0.045 | 0.908 | 0.898 | 0.970 | 0.925 | 0.031 | 0.712 | 0.799 | 0.950 | 0.819 |
| ACUTE mid act Beta | 0.025 | 0.910 | 0.922 | 0.972 | 0.934 | 0.037 | 0.710 | 0.798 | 0.950 | 0.818 |
| ACUTE late act | 0.077 | 0.849 | 0.880 | 0.968 | 0.898 | 0.050 | 0.691 | 0.799 | 0.950 | 0.812 |
| ACUTE late act Platt | 0.030 | 0.903 | 0.913 | 0.971 | 0.929 | 0.036 | 0.693 | 0.799 | 0.950 | 0.813 |
| ACUTE late act Isotonic | 0.032 | 0.907 | 0.905 | 0.972 | 0.928 | 0.037 | 0.700 | 0.799 | 0.950 | 0.815 |
| ACUTE late act Beta | 0.029 | 0.903 | 0.912 | 0.971 | 0.929 | 0.036 | 0.693 | 0.799 | 0.950 | 0.813 |

*Table 17.* Results on the APIGen test subset (left), and on the SCITLDR dev set (right) on SmolLM3-3B. Lower smECE is better, while higher AUC-EURO is better.

| | APIGen | | | | | SCITLDR | | | | |
|---|---|---|---|---|---|---|---|---|---|---|
| | | AUC-EURO (↑) | | | | | AUC-EURO (↑) | | | |
| estimator | smECE (↓) | low | med | high | all | smECE (↓) | low | med | high | all |
| Raw Conf | 0.261 | 0.465 | 0.471 | 0.594 | 0.510 | 0.146 | 0.413 | 0.760 | 0.939 | 0.701 |
| HRE | 0.028 | 0.941 | 0.799 | 0.718 | 0.820 | 0.088 | 0.637 | 0.763 | 0.940 | 0.779 |
| NWKR | 0.043 | 0.938 | 0.760 | 0.688 | 0.797 | 0.083 | 0.641 | 0.763 | 0.940 | 0.780 |
| Platt | 0.025 | 0.938 | 0.753 | 0.669 | 0.788 | 0.082 | 0.643 | 0.763 | 0.940 | 0.781 |
| Isotonic | 0.021 | 0.938 | 0.753 | 0.688 | 0.795 | 0.082 | 0.637 | 0.763 | 0.940 | 0.778 |
| Beta | 0.036 | 0.938 | 0.753 | 0.679 | 0.791 | 0.083 | 0.641 | 0.761 | 0.940 | 0.779 |
| ACUTE cosine | 0.031 | 0.946 | 0.817 | 0.757 | 0.841 | 0.089 | 0.634 | 0.760 | 0.940 | 0.776 |
| ACUTE cosine Platt | 0.025 | 0.945 | 0.818 | 0.758 | 0.841 | 0.057 | 0.658 | 0.761 | 0.940 | 0.785 |
| ACUTE cosine Isotonic | 0.029 | 0.946 | 0.817 | 0.749 | 0.838 | 0.063 | 0.650 | 0.763 | 0.940 | 0.783 |
| ACUTE cosine Beta | 0.031 | 0.945 | 0.819 | 0.753 | 0.840 | 0.056 | 0.658 | 0.762 | 0.940 | 0.785 |
| ACUTE pca10 | 0.025 | 0.946 | 0.834 | 0.813 | 0.865 | 0.092 | 0.637 | 0.762 | 0.940 | 0.778 |
| ACUTE pca10 Platt | 0.029 | 0.947 | 0.833 | 0.811 | 0.865 | 0.056 | 0.666 | 0.762 | 0.940 | 0.788 |
| ACUTE pca10 Isotonic | 0.051 | 0.940 | 0.835 | 0.798 | 0.859 | 0.062 | 0.658 | 0.760 | 0.938 | 0.784 |
| ACUTE pca10 Beta | 0.037 | 0.945 | 0.832 | 0.809 | 0.863 | 0.060 | 0.665 | 0.761 | 0.940 | 0.787 |
| ACUTE pca20 | 0.027 | 0.947 | 0.843 | 0.812 | 0.868 | 0.092 | 0.653 | 0.765 | 0.940 | 0.785 |
| ACUTE pca20 Platt | 0.026 | 0.948 | 0.844 | 0.812 | 0.869 | 0.052 | 0.678 | 0.763 | 0.940 | 0.793 |
| ACUTE pca20 Isotonic | 0.043 | 0.946 | 0.840 | 0.802 | 0.864 | 0.052 | 0.668 | 0.763 | 0.940 | 0.789 |
| ACUTE pca20 Beta | 0.036 | 0.947 | 0.844 | 0.806 | 0.866 | 0.052 | 0.678 | 0.763 | 0.940 | 0.793 |
| ACUTE early act | 0.047 | 0.942 | 0.825 | 0.784 | 0.851 | 0.089 | 0.652 | 0.763 | 0.940 | 0.784 |
| ACUTE early act Platt | 0.032 | 0.945 | 0.834 | 0.788 | 0.857 | 0.055 | 0.675 | 0.763 | 0.940 | 0.791 |
| ACUTE early act Isotonic | 0.052 | 0.938 | 0.832 | 0.785 | 0.852 | 0.059 | 0.672 | 0.763 | 0.940 | 0.790 |
| ACUTE early act Beta | 0.033 | 0.944 | 0.834 | 0.788 | 0.856 | 0.056 | 0.674 | 0.763 | 0.940 | 0.791 |
| ACUTE mid act | 0.057 | 0.943 | 0.844 | 0.821 | 0.870 | 0.088 | 0.653 | 0.763 | 0.940 | 0.784 |
| ACUTE mid act Platt | 0.026 | 0.950 | 0.854 | 0.830 | 0.879 | 0.053 | 0.676 | 0.762 | 0.940 | 0.792 |
| ACUTE mid act Isotonic | 0.037 | 0.945 | 0.852 | 0.816 | 0.872 | 0.056 | 0.680 | 0.763 | 0.936 | 0.792 |
| ACUTE mid act Beta | 0.037 | 0.948 | 0.854 | 0.825 | 0.876 | 0.053 | 0.676 | 0.763 | 0.940 | 0.792 |
| ACUTE late act | 0.063 | 0.943 | 0.851 | 0.822 | 0.872 | 0.087 | 0.654 | 0.764 | 0.940 | 0.784 |
| ACUTE late act Platt | 0.027 | 0.951 | 0.856 | 0.831 | 0.880 | 0.061 | 0.675 | 0.763 | 0.940 | 0.791 |
| ACUTE late act Isotonic | 0.050 | 0.944 | 0.854 | 0.813 | 0.871 | 0.057 | 0.666 | 0.763 | 0.936 | 0.787 |
| ACUTE late act Beta | 0.038 | 0.951 | 0.857 | 0.821 | 0.877 | 0.063 | 0.675 | 0.762 | 0.939 | 0.791 |

*Table 18.* Results on the APIGen test subset (left), and on the SCITLDR dev set (right) on Qwen3-30B-A3B-Instruct-2507. Lower smECE is better, while higher AUC-EURO is better.

| estimator | | APIGen | | | | | SCITLDR | | | | |
|---|---|---|---|---|---|---|---|---|---|---|---|
| | smECE (↓) | \multicolumn AUC-EURO (↑) | | | | smECE (↓) | AUC-EURO (↑) | | | |
| | | low | med | high | all | | low | med | high | all |
| Raw Conf | 0.222 | 0.305 | 0.520 | 0.766 | 0.528 | 0.152 | 0.366 | 0.721 | 0.925 | 0.667 |
| HRE | 0.023 | 0.841 | 0.717 | 0.805 | 0.788 | 0.081 | 0.665 | 0.721 | 0.925 | 0.769 |
| NWKR | 0.029 | 0.841 | 0.701 | 0.799 | 0.781 | 0.082 | 0.667 | 0.721 | 0.925 | 0.770 |
| Platt | 0.031 | 0.845 | 0.711 | 0.798 | 0.785 | 0.079 | 0.668 | 0.721 | 0.925 | 0.771 |
| Isotonic | 0.017 | 0.843 | 0.696 | 0.797 | 0.780 | 0.079 | 0.669 | 0.721 | 0.925 | 0.771 |
| Beta | 0.024 | 0.840 | 0.698 | 0.795 | 0.778 | 0.079 | 0.668 | 0.721 | 0.925 | 0.771 |
| ACUTE cosine | 0.035 | 0.876 | 0.772 | 0.827 | 0.826 | 0.085 | 0.669 | 0.717 | 0.925 | 0.769 |
| ACUTE cosine Platt | 0.035 | 0.874 | 0.773 | 0.828 | 0.825 | 0.072 | 0.678 | 0.721 | 0.925 | 0.774 |
| ACUTE cosine Isotonic | 0.043 | 0.873 | 0.769 | 0.821 | 0.822 | 0.078 | 0.673 | 0.718 | 0.924 | 0.771 |
| ACUTE cosine Beta | 0.036 | 0.874 | 0.773 | 0.826 | 0.825 | 0.073 | 0.678 | 0.721 | 0.925 | 0.774 |
| ACUTE pca10 | 0.036 | 0.906 | 0.821 | 0.863 | 0.864 | 0.089 | 0.685 | 0.723 | 0.925 | 0.777 |
| ACUTE pca10 Platt | 0.030 | 0.909 | 0.822 | 0.861 | 0.865 | 0.074 | 0.694 | 0.722 | 0.925 | 0.779 |
| ACUTE pca10 Isotonic | 0.041 | 0.903 | 0.821 | 0.857 | 0.860 | 0.075 | 0.686 | 0.723 | 0.925 | 0.777 |
| ACUTE pca10 Beta | 0.032 | 0.908 | 0.822 | 0.861 | 0.864 | 0.074 | 0.694 | 0.722 | 0.925 | 0.779 |
| ACUTE pca20 | 0.054 | 0.915 | 0.837 | 0.866 | 0.873 | 0.091 | 0.688 | 0.723 | 0.925 | 0.778 |
| ACUTE pca20 Platt | 0.029 | 0.922 | 0.843 | 0.868 | 0.878 | 0.073 | 0.698 | 0.723 | 0.925 | 0.782 |
| ACUTE pca20 Isotonic | 0.037 | 0.917 | 0.838 | 0.864 | 0.874 | 0.072 | 0.692 | 0.724 | 0.925 | 0.780 |
| ACUTE pca20 Beta | 0.030 | 0.922 | 0.843 | 0.868 | 0.878 | 0.072 | 0.699 | 0.723 | 0.925 | 0.782 |
| ACUTE early act | 0.050 | 0.901 | 0.813 | 0.852 | 0.855 | 0.086 | 0.686 | 0.722 | 0.925 | 0.777 |
| ACUTE mid act | 0.063 | 0.907 | 0.834 | 0.871 | 0.871 | 0.084 | 0.689 | 0.722 | 0.925 | 0.778 |
| ACUTE late act | 0.068 | 0.907 | 0.834 | 0.870 | 0.870 | 0.084 | 0.681 | 0.722 | 0.925 | 0.775 |

*Table 19.* Results on the APIGen test subset (left), and on the SCITLDR dev set (right) averaged across all 7 LLMs we tested. Lower smECE is better, while higher AUC-EURO is better.

