# OpenReview forum: "The ACUTE Protocol: Operationalizing Language Model Activations for Better Calibration, Utility, and Trust"
_ICML.cc/2026/Conference — ICML 2026 regular_

### Official Review · Reviewer_PP1i · 2026-03-13

**Soundness:** 3
**Presentation:** 3
**Significance:** 4
**Originality:** 3
**Overall Recommendation:** 5
**Confidence:** 3

**Summary:**

The papers tackles a real shortcoming of expected calibration error (ECE) as the dominant metric for LLM confidence estimation. ECE can be gamed and has no discriminative power when a trivial confidence estimator outputs the empirical base rate always. To mitigate it, the paper proposes expected utility renormalized by the oracle (EURO) as a decision-theoretic metric parameterize by a risk level that evaluates whether a confidence estimator correctly routes decisions by penalizing trusting incorrect outputs and rejecting correct ones. In addition, the paper yet makes another remote contribution that proposes an activation-based confidence estimator, termed ACE, that train a random forest classifier on pooled activations of LLMs. ACE is evaluated on three tasks including multiple-choice QA, tool-calling and summarization over 6 LLMs and is empirically shown better than raw confidence, and other learning-based confidence estimators including histogram regression and kernel regression.

**Compliance With Llm Reviewing Policy:**

Affirmed.

**Final Justification:**

The rebuttal addressed all my concerns except one which is to ask for an empirical result on a more popularly-adopted uncertainty/confidence estimator on the open-ended generation task. Please see the details in my rebuttal acknowledgement. I think this is an important work to look at a metric for practical deployment beyond relying on calibration alone.

**Key Questions For Authors:**

The paper is a good read and there are a few concrete steps that can improve the evaluation of the paper. (1) Add the missing ablation on with and without raw confidence as the inputs to the classifiers. (2) Add a post-hoc calibration over ACE and see if the outcome gain is additive (3) Add some popular estimators from UQ for LLMs literature (4) discuss the data splits of experiments that involves training (5) soften the tone for contradictory/over-claimed sentences. (6) Is EURO strictly bounded in [0,1] after normalization? How is the "anti‑oracle" defined formally? (6) Discuss connection to selective prediction literature and its AURC metric.

**Limitations:**

yes

**Strengths And Weaknesses:**

The EURO metric is well-motivated and fills a genuine gap. The concrete failure mode of ECE is a compelling illustration and the writing of the technical section is built progressively and easy to follow. The proposed metric is grounded by the expected net utility at the Bayes-optimal trust threshold, normalized by the oracle's achievable utility, and the EURO at each risk level naturally admits a curve for a holistic evaluation without picking a operating point which might be task dependent. The sample complexity of the proposed ACE is noteworthy: it's incredible to see that ACE with 25 training samples performs better than baselines trained with 1000 samples. The empirical layer-wise analysis across all six LLMs shows a consistent trend that might shed lights to follow-up works in how LLMs encode confidence.

Given the utility definition and the nature of abstention likely incorrect answers, I think selective prediction literature as well as its governing metric like area under the risk-coverage curve (AURC) shall be included in the discussion. There is one missing ablation which is on the necessity of auxiliary raw confidence as inputs to ACE classifiers. On APIGen, ACE variants improve AUC-EURO but also have higher smECE. It'll be great to discuss when high decision utility justifies worse calibration, and whether an additional step of post-hoc calibration of ACE outputs would preserve AUC-EURO while reducing smECE. The $\tau=u_{ca}$ decision rule is Bayes-optimal when the confidence estimator's outputs equal true correctness probabilities. In practice, it seems it will not be the case always, but the paper does not discuss what happens to the EURO metric under this scenario. A simple check is to apply post-hoc calibration and verify whether EURO improves. In Section 5.1, the statement "ACE always outperforms baselines on AUC-EURO" is slightly overclaim as it directly contracts with Table 1's SCITLDR results. Similarly, Section 5.4 it says "These trends are consistent with all LLMs we tested" but without showing data for all 6 LLMs. Last but not least, it seems the authors conveniently omit many confidence/uncertainty estimators regardless calibration, such as a rich body of Monte-Carlo sampling-based uncertainty estimates. It seems the EURO metric is estimator-agnostic so any scoring function can be evaluated. Given that, the baseline set feel narrow.

---

> ### Author Rebuttal · Authors · 2026-03-31
>
> Thank you for the comprehensive review. We’re glad you find our work well-motivated, filling a genuine gap, our metric (EURO) grounded and holistic, and our method’s (ACE) sample efficiency noteworthy. Below we address your concerns.
>
> **Q1:** For space reasons, we show the delta of adding the confidence feature for pca10 and pca20 across all 3 tasks (MMLU averaged across all 57 subtasks). These are averaged across the 6 LLMs we test and result in a negligibly positive difference.
>
> | Estimator | Task Name | smECE | AUC EURO (all) |
> | --- | --- | --- | --- |
> | pca10 | mmlu | 0.0 | +0.002 |
> | pca20 | mmlu | 0.0 | +0.001 |
> | pca10 | apigen | 0.0 | 0.0 |
> | pca20 | apigen | +0.002 | 0.0 |
> | pca10 | scitldr | 0.0 | 0.0 |
> | pca20 | scitldr| +0.001 | 0.0 |
>
> **Q2:** This is a great suggestion. We incorporate three post-hoc calibration methods here: platt scaling, isotonic regression, and beta calibration both as baselines and post-hoc calibrators for the RF outputs for ACE methods. Baselines perform comparably to NWKR on both AUC EURO and smECE. Below we include a table of results for ACE pca20 posthoc calibrated on APIGen, averaged across the 6 LLMs, indicating that posthoc calibration slightly improves AUC EURO, while reducing smECE by a factor of 2, so *results are additive*. In the final version, we will include all ACE methods post-hoc calibrated across all tasks and models. Note for this, we reran our experiments, splitting our training set of 1000 examples into an 800 example training set and a 200 example calibration set, so these numbers vary slightly from Table 1 in the paper, which trained on all 1000 examples.
>
> | Estimator | PostHoc | smECE | AUC EURO (all) |
> | --- | --- | --- | --- |
> | NWKR | none | 0.026 | 0.778 |
> | platt | none | 0.032 | 0.785 |
> | isotonic | none | 0.016 | 0.777 |
> | beta | none | 0.021 | 0.776 |
> | ACE pca20 | none | 0.058 | 0.873 |
> | ACE pca20 | platt | 0.029 | 0.880 |
> | ACE pca20 | isotonic | 0.036 | 0.875 |
> | ACE pca20 | beta | 0.029 | 0.880 |
>
> **Q3:** We added 3 post-hoc calibration methods as baselines: platt scaling, isotonic regression, and beta calibration as mentioned in the previous section. They perform comparably to the NWKR baseline.
>
> **Q4:** For MMLU, for each task we split into training and evaluation sets. If the task had less than 200 examples, we used 50 for training, less than 400 examples, 100 for training, and otherwise 200 for training. All other examples were put in the evaluation set. For APIGen we randomly assigned 1000 examples for training and 1000 for evaluation. For SCITLDR, we use the training set (1992 samples) for training and the development set (618 samples) for evaluation. We will include this in the final version in section 4 + appendix. When including the post-hoc calibration, we split each training set into 80% training and 20% calibration.
>
> **Q5:** Our SCITLDR results are *significant* for 5 of the 6 LLMs. See the response to Reviewer B8ZC under W2. We also include analysis varying the rouge-L threshold, which show consistent results. See our response to Reviewer B8ZC under Q4.
>
> **Q6a,b:** Yes EURO is strictly bounded [0, 1]. The anti-oracle (AO) is a system that does the opposite of what the oracle does (only gets false positives and negatives, so for every correct generation, AO abstains and for every incorrect one AO trusts; see lines 176-177).
>
> **Q6c:** From our understanding, a selective prediction system can decide to output a class label or abstain from it and is governed by a selective function. In our setting, our selective function is a combination of the confidence estimator and the minimum bayes risk-based optimal policy. The selective classifier tries to minimize the selective risk at a given coverage value. In our setting, we define the risk level according to the normalized net utility of correctly abstaining (uca) based on the bayes-optimal policy, and evaluate EURO according to that risk level. Given this, the risk-coverage curve defined in selective prediction literature and our uca vs. EURO curve are different.
>
> **Other things discussed in strengths and weaknesses:** Worse calibration for better utility is especially valuable if the calibration is uninformative (e.g. if an estimator gets low calibration because it only predicts the base rate, like HRE+NWKR for SCITLDR). Even though ACE only gets slightly higher AUC EURO as compared to HRE/NWKR, the spread of probabilities is better for decision-making utility (see Figure 3, bottom row). Note EURO does not exclusively measure decision-making utility (it incorporates calibration too).
>
> Thank you for helping us strengthen our work. Anything added here, we will add to the final version, including experiments with Qwen3-30B-A3B-Instruct-2507, a large MoE model, following reviewer B8ZC's suggestion. If you have any other questions or concerns, we’d be happy to answer them. We would appreciate it if you could update your review taking our response into account.

---

> > ### Author Rebuttal · Reviewer_PP1i · 2026-04-02
> >
> > Thank you for spending time resolving my concerns.
> >
> > Q1: I appreciate the ablation and please include in the revision if the finalized method still keeps the confidence feature.
> >
> > Q2: Excellent!
> >
> > Q3: I appreciate the post-hoc calibration baselines, but they are orthogonal to the MC-based UQ that I mentioned. Some examples could include self consistency, semantic entropy or any forms of sequence-level agreement UQ methods. Raw-confidence-based estimators suffer from length-bias for open-ended generation tasks beyond multiple-choice QA tasks, so a non raw-confidence-based estimator would be a nice-to-have. The community hasn't really converged to a standard for how to conduct UQ for open-ended LLM generation, but empirically many agreement-based UQ methods on the sequence -level are shown better than raw-confidence-based ones.
> >
> > Q4: Clear description.
> >
> > Q5: That confirms that the language for some experimental claims needs to be soften.
> >
> > Q6: The distinction between the two curves to dismiss selective prediction literature are valid, and please include this discussion explicitly in the paper text.
> >
> > The only remaining question is Q3. Based on the resolution of my other concerns (especially Q2), I raise my score to 5.

---

> > > ### Author Response · Authors · 2026-04-05
> > >
> > > Thank you for your constructive feedback, engagement, and incorporation of our feedback into your review including raising your score.
> > >
> > > Q1: We plan to put it in the final version. The numbers presented throughout are with the confidence feature.
> > >
> > > Q3: Our setup doesn't lend itself to being able to compare to these baselines directly at the moment. We use greedy decoding to generate a single candidate answer per example. From our understanding of some of those MC-based UQ estimators, they require k candidate answers per instance/example and without those the clusters cant easily be measured. In future work, we could generate many candidates per instance (for the multi-token generation settings) and compare our methods versus some of these MC-based UQ choices like semantic entropy. We will put a note in the paper saying why we couldn't directly compare to these methods.
> > >
> > > Q5: We agree; we will soften the language in the final version.
> > > Q6: We will put this in the final version too.

---

### Official Review · Reviewer_Xvgf · 2026-03-17

**Soundness:** 4
**Presentation:** 4
**Significance:** 4
**Originality:** 4
**Overall Recommendation:** 6
**Confidence:** 4

**Summary:**

This paper introduces "Expected utility renormalized by the oracle" (EURO) as a novel method to measure risk-aware calibration error which requires only a choice of risk threshold, as well as the "Activation-based confidence estimation" (ACE) protocol as a recalibration technique using activations from LLM layers as inputs to a practical featurization pipeline and a confidence classifier. The authors explain EURO in sufficient detail with sufficient background motivation and ACE through extensive evaluations of multiple LLMs on two datasets.

**Compliance With Llm Reviewing Policy:**

Affirmed.

**Key Questions For Authors:**

No further questions from my side.

**Limitations:**

yes

**Strengths And Weaknesses:**

Strengths:
- Soundness: The EURO calibration measure is technically sound and sets appropriate context using a toy example. The toy example showcases weaknesses of ECE and smECE where they fail to capture nuances and differences between a discrete oracle, a probabilistic oracle, a constant base rate, and a noisy base rate with uniform distribution over a large range.
- Presentation: The paper is written in a highly engaging style, with mathematical derivations and commentary that are easy to follow.
- Significance: The EURO calibration measure addresses the need for introducing a tuneable task-specific risk threshold, which is important to control in production applications with low or high tolerance towards the trust <-> abstention tradeoff. ACE demonstrates how recalibration with an estimator has greater impact under the risk-aware EURO measure compared to other standard baseline calibration measures and recalibration methods.
- Originality: The ACE protocol recalibrates LLM confidence by extracting activations from LLM layers, processing them through a featurization step and training a classifier, which decides whether to trust the output or abstain at the chosen risk threshold. The idea of incorporating activations into confidence recalibration is not new and requires white-box access to the LLM. However, the authors devised a rich pipeline for processing the activations and tested multiple distinct featurization techniques, which grounds this contribution into practical settings.

Weaknesses:
- There were no major weaknesses that I could identify in the proposed methods.

---

> ### Author Rebuttal · Authors · 2026-03-31
>
> Thank you for your comprehensive review. We’re really glad you enjoyed our paper and found our paper technically sound, highly engaging, significant, original, and grounded in practical settings without any major weaknesses. If you have any questions or additional feedback, we’d be happy to answer them.

---

> > ### Author Rebuttal · Reviewer_Xvgf · 2026-04-03
> >
> > No further questions

---

> > > ### Author Response · Authors · 2026-04-05
> > >
> > > Thank you for your comprehensive review and glad you enjoyed the paper!

---

### Official Review · Reviewer_kMYp · 2026-03-21

**Soundness:** 2
**Presentation:** 2
**Significance:** 2
**Originality:** 2
**Overall Recommendation:** 4
**Confidence:** 3

**Summary:**

This paper discussed two problems in LLM trustworthiness: the inadequacy of existing calibration metrics (ECE, smECE) for decision-making, and the lack of practical confidence estimation methods that work well at high risk levels. The authors propose EURO, a decision-theoretic metric that incorporates task risk level and penalizes uninformative estimators that ECE cannot distinguish from oracles. They also introduce ACE, an activation-based confidence estimation framework using random forests trained on mean-pooled activations, cosine similarities, or PCA-reduced representations. Experiments across 6 models and 3 tasks (MMLU, APIGen, SCITLDR) show ACE consistently outperforms recalibration baselines on AUC-EURO while maintaining competitive smECE.

**Compliance With Llm Reviewing Policy:**

Affirmed.

**Final Justification:**

I think my main concerns have been addressed. The authors showed the significance test that justify their reject.

**Key Questions For Authors:**

- Q1. Are the gains statistically significant? The improvements on SCITLDR are very small, often below 0.01 AUC-EURO, and the paper does not report significance tests or variance estimates. Could the authors provide confidence intervals or statistical tests for the main results, especially on SCITLDR?

- Q2. How practical is the method under black-box access? The framework requires access to intermediate activations, but many high-stakes systems rely on black-box API models. Could the authors clarify whether ACE can be adapted to settings with only outputs or token probabilities, or better discuss the practical scope of the method under this limitation?

- Q3. How sensitive are the results to the SCITLDR threshold? The choice of ROUGE-L ≥ 0.3 is central to both training and evaluation, yet it is not validated. Could the authors provide a sensitivity analysis under different thresholds, or evidence that this binarization aligns with human judgments?

**Limitations:**

- Random forests may introduce calibration bias.
The paper uses random forests as the main confidence estimator, but the justification is fairly limited. Since random forests often produce overconfident probability estimates, it would be helpful to better verify whether this behavior affects ACE in the activation-space setting considered here. In particular, the paper does not provide a direct calibration analysis of the random forest outputs themselves.

- Averaging across models and tasks may hide heterogeneity.
Table 1 reports results averaged across all six models and, for MMLU, across all 57 subtasks. While this gives a useful summary, it may also mask important variation across model sizes or task domains. A more detailed per-model or per-subtask breakdown would make it easier to judge how consistent the gains really are.

- The intrinsic dimension analysis is interesting but not very actionable.
The PCA-based analysis in Section 5.4 provides a useful descriptive picture of how representation complexity changes across layers, but it is not clearly tied back to ACE’s design choices. For example, it does not explain the selected number of PCA components, why later layers work better, or how practitioners should use this analysis when applying ACE to new models. As a result, the section feels more descriptive than operational.

**Strengths And Weaknesses:**

Strengths

- S1. EURO addresses a clear and important gap.
The paper makes a strong case that existing calibration metrics such as ECE and smECE can rate a base-rate estimator as perfect, failing to distinguish an oracle from an uninformative predictor. EURO addresses this issue in a principled way through a Bayes-risk perspective, with a single interpretable parameter ​u_{ca}

- S2. Strong sample efficiency.
A particularly compelling result is that ACE-based estimators trained on only 25 examples outperform stronger baselines trained on the full 1000-example set. This is highly relevant in practice, where labeled calibration data is often limited.

- S3. Broad empirical coverage.
The evaluation spans three different task types and six models from four families, which gives meaningful evidence of generality. The consistent layer-wise trends across models are also an interesting finding in their own right.

- S4. Clear presentation and solid theory.
The paper is well written, and the theoretical development of EURO is clean and easy to follow. The reduction from four reward parameters to a single u_{ca} makes the metric both principled and interpretable.

Weaknesses

- W1. Some gains are very small and not statistically tested.
On SCITLDR in particular, the differences between ACE and the baselines are quite small, and the paper does not report significance tests or variance estimates. This makes it difficult to judge whether the reported improvements are reliable.

- W2. The SCITLDR correctness threshold is not well justified.
The use of ROUGE-L ≥0.3 to define correctness is an important design choice, but it is only lightly discussed. Since the framework depends on this binary label, the paper would benefit from a sensitivity analysis over the threshold.

- W3. Access to model internals limits applicability.
ACE requires intermediate activations from the target model, which is feasible for open-weight models but not for most frontier API-based systems. This practical limitation is important and not sufficiently discussed.

- W4. Comparison with MICE is incomplete.
The paper positions ACE as a cheaper approximation to MICE, but does not provide a direct empirical comparison on shared settings. A more explicit comparison would help clarify the paper’s novelty and empirical advantage.

- W5. The EURO risk bins seem somewhat arbitrary.
The low/medium/high partitions of u_{ca} are introduced without much justification. It would be helpful to explain why these boundaries are meaningful or show that the results are robust to alternative choices.

---

> ### Author Rebuttal · Authors · 2026-03-31
>
> Thanks for the constructive review. We’re encouraged that you think our metric, EURO, addresses an important gap with solid theory and our methods (ACE) have strong sample efficiency with broad empirical coverage. Below we address your concerns.
>
> W1+Q1: We perform permutation tests to test the difference in AUC EURO for our ACE methods as compared to the strongest baseline (nwkr). For space, we describe the results, but will update the paper with all of the tests and associated p-values. On APIGen, p-values are less than 0.01 for each model between the strongest ACE methods and the NWKR baseline. We added post-hoc calibration, following the suggestion of reviewer PP1i both as baselines and to recalibrate ace outputs. After beta recalibration, AUC-EURO on SCITLDR improved slightly and for 5 of the 6 models, the AUC-EURO difference is significant at 0.05 (p=0.04, 0.002, 0.01, 0.008, 0.06, and 0.002 for the 6 models). Only phi-4 is not significant. Additionally, AUC-EURO understates the impact of ACE; the reliability diagrams in Figure 3 show a larger spread of probability mass (baselines masquerade as base rate estimators). See discussion with Reviewer B8ZC under W2 and W3+Q2.
>
> W2+Q3: SCITLDR is a very challenging task, so the performance at higher rouge-L levels is very low, leading to never trusting the model output to be nearly optimal. See discussion with Reviewer B8ZC under W4 and Q2.
>
> W3+Q2: We will include requiring access to model activations in our limitations. Despite this, we don’t view this as a major limitation; the developers of a language model will always have access to its internals. In this competitive landscape, LLM developers are incentivized to improve user trust in their system, so incorporating ACE would be beneficial. Both Anthropic and Google have used model internals on their closed-source models for a variety of use cases [1,2]. ACE could be minimally adapted to use full or partial logit outputs, but we suspect that performance could suffer given features are taken from the top of the network.
>
> W4: In appendix C, we discuss how ACE cosine is nearly identical to MICE, just more efficient by using the underlying LM as the BERTScore LM rather than having an auxiliary one. MICE is impractically slow for deployment (thousands of times slower than ACE; see reviewer B8ZC Q3 for ACE’s speed).
>
> W5: The bayes-optimal policy is to trust an output if p > u_{ca}, so we stratified into 3 equal thirds, signifying low-risk (a probability of only at most 0.33 is needed to trust), medium-risk (probability between 0.33 and 0.67 needed to trust), and high-risk (need at least a probability of 0.67 to trust). For a given task, it's hard to pinpoint the specific risk associated with the task; it's easier to give a range and measure the associated AUC-EURO over that range. EURO values are consistent with small perturbations in u_{ca}.
>
> L1: We are confused by what you mean here. We take the output of the trained RF as the new calibrated confidence, so our measures of smECE and EURO are based on RF output probabilities. Following the suggestion of Reviewer PP1i (response under Q2), we added three additional posthoc calibration baselines and post-hoc calibrated our random forests. Post-hoc calibration improves AUC EURO slightly, but reduces smECE often to below NWKR levels across models and tasks.
>
> L2: Here are the per model results presenting the difference between AUC-EURO (all) for pca20 vs. NWKR per model across all tasks. Positive numbers indicate gains that ACE has over NWKR. Results are consistent across models with the larger models having greater improvement over the baseline. We will add the full tables with all methods per model to the appendix including the new post-hoc calibration baselines and additions (see Reviewer PP1i Q2).
>
> | LLM | mmlu | apigen | scitldr (0.3) |
> | --- | --- | --- | --- |
> | SmolLM3 | 0.00 | 0.08 | 0.02 |
> | gemma3-4b | 0.02 | 0.08 | 0.01 |
> | gemma3-12b | 0.03 | 0.12 | 0.01 |
> | phi-4 | 0.04 | 0.16 | 0.01 |
> | qwen3-4b | 0.03 | 0.07 | 0.01 |
> | qwen3-14b | 0.01 | 0.07 | 0.01 |
>
> L3: We used the PCA analysis to better motivate our choice of dimensionality reduction for our ACE pca10 and pca20 methods. Our intrinsic dimensionality analysis helps show that information is not lost and that later layers have more complex representations (i.e. they cannot be compressed as tightly and thus the number of directions needed to explain X% of the variance is greater at later layers than earlier ones). This helps explain why later layers have more extractable confidence signals.
>
> Thank you for helping us strengthen our work. We will add all of this to the final version. If you have any other questions or concerns, we’d be happy to answer them. We would appreciate it if you could update your review taking our response into account.
>
> [1] Building Production-Ready Probes for Gemini (Kramar et al. 2026)
>
> [2] Claude Sonnet 4.5 System Card (Sec7.6 Whitebox interpretability investigations)

---

> > ### Author Rebuttal · Reviewer_kMYp · 2026-04-02
> >
> > Thank the author for your reply, I will raise my score.

---

> > > ### Author Response · Authors · 2026-04-05
> > >
> > > Thank you for your constructive feedback, engagement, and incorporation of our feedback into your review including raising your score.

---

### Official Review · Reviewer_B8ZC · 2026-03-21

**Soundness:** 2
**Presentation:** 2
**Significance:** 3
**Originality:** 2
**Overall Recommendation:** 4
**Confidence:** 3

**Summary:**

This study makes two contributions. First, it proposes EURO, a new metric for evaluating confidence estimators that considers both calibration and decision-making utilities. The EURO fixes known problems with ECE and smECE, which cannot distinguish a perfect oracle from a useless baserate predictor. Second, it proposes ACE, a method that uses the internal activations of LLMs to train a random forest classifier that predicts whether a model output is correct. The predicted probability of the classifier served as a recalibrated confidence score. The experiments covered three tasks (MMLU, APIGen, and SCITLDR) and six models from four families, showing that ACE improves AUC-EURO while keeping the calibration error low.

**Compliance With Llm Reviewing Policy:**

Affirmed.

**Final Justification:**

The rebuttal addressed my main concerns: significance tests (W3), the MoE experiment (W4), and the noise ablation (W1) collectively strengthen the empirical claims. I raise my score to 4, contingent on the final version including all the promised additions.

**Key Questions For Authors:**

1. Can you add at least one baseline with richer input than a single scalar?

2. Can you show per-model results with standard deviations and run significance tests?

3. What is the actual wall-clock overhead of the ACE pipeline per query?

4. How do SCITLDR results change with rouge-L thresholds of 0.2, 0.4, and 0.5?

**Limitations:**

The paper includes an Impact Statement that correctly notes EURO and ACE can help improve trust in LLM outputs, and acknowledges that the decision-theoretic framework is not a replacement for careful data analysis and system design.

However, the paper does not acknowledge that ACE is limited to open-weight models with accessible activations, excluding widely deployed closed-source systems.

Technical limitations such as the narrow model scale range tested (3B–14B), the dependence on labeled correctness data for training, and the sensitivity of the generation-task setup to the binarization threshold are not mentioned.

**Strengths And Weaknesses:**

S1:

The EURO metric fills a real gap. The paper convincingly shows (Figure 1) that ECE and smECE give the same score to a perfect oracle and a useless baserate predictor — this is a serious problem for anyone using these metrics to evaluate confidence estimators. EURO fixes this by incorporating the risk level into the evaluation. The metric design is clean, and the derivation is sound. This is a useful standalone contribution.

S2:

Strong results on APIGen and MMLU. On APIGen, ACE pca20 reaches 0.88 AUC-EURO versus 0.78 for the best baseline — a +0.10 gap averaged over 6 models, which is substantial.

S3:

Impressive sample efficiency. ACE trained on just 25 examples outperforms baselines trained on 1000 examples.

W1:

The baselines are too weak for a fair comparison. HRE and NWKR only take a single number (raw confidence) as input. ACE uses thousands of activation features plus the raw confidence. A random forest with rich features beating a univariate method is expected — this does not tell us whether activations are specifically useful for confidence estimation, or whether any high-dimensional features would do.

W2:

SCITLDR results are essentially flat. ACE gets 0.78 versus 0.77 for baselines — a 0.01 difference that is likely noise.

W3:

No significance tests. Table 1 averages over 6 models (and 57 MMLU subtasks) but shows no standard deviations, confidence intervals, or p-values. Per-model breakdowns are missing from the main paper. We cannot tell if ACE consistently helps all models or just some.

W4:

Only works with open-weight models. ACE needs access to hidden-layer activations, so it cannot be applied to closed-source models such as GPT-4 or Claude. The paper describes ACE as "general-purpose" but does not acknowledge this limitation. All tested models are also in the 3B-14B range — generalization to 70B+ or mixture-of-experts models remains untested.

---

> ### Author Rebuttal · Authors · 2026-03-31
>
> Thanks for the detailed review. We’re glad you find our metric a useful standalone contribution, our results across 2 tasks substantial, and our sample efficiency impressive. Below we address your concerns:
>
> W1+Q1: We currently evaluate 3 baselines (raw confidences, hre, nwkr). We add 3 more post-hoc calibration methods following Reviewer PP1i’s suggestion under Q2; these perform comparably to nwkr. Most baselines calibrate raw confidences directly. To test whether any multi-variate vectors help with confidence estimation, we construct a noise matrix by sampling from a multi-variate gaussian parametrized by each feature’s mean and variance. We interpolate between the original features and the noise matrix on both training and test sets and evaluate ACE pca20 via smECE and AUC EURO averaged across all LLMs on APIGen. We include the base rate estimator (one that just predicts training set accuracy for all inputs). Results hint that the activations contain relevant signals for confidence estimation and that performance degrades to the base rate estimator with more noise.
>
> | noise | smECE | AUC EURO (all) |
> | --- | --- | --- |
> | 0% | 0.05 | 0.873 |
> | 25% | 0.06 | 0.867 |
> | 50% | 0.06 | 0.850 |
> | 75% | 0.05 | 0.814 |
> | 100% | 0.05 | 0.778 |
> | baserate | 0.01 | 0.777 |
>
> W2: AUC-EURO alone undersells the impact of ACE on confidence estimation. The reliability diagrams in the bottom row of Figure 3 show that both ACE systems spread out probability mass much more than baselines. Baselines resemble the base rate estimator. Adding beta calibration post-hoc to pca20 improved AUC-EURO slightly and permutation tests for SCITLDR (rouge-L threshold=0.3, 1000 resamples) shows that the difference between AUC-EURO for pca20 and nwkr is **statistically significant** at alpha=0.05 for 5 of the 6 LLMs: gemma3-12b (p=0.038), gemma3-4b (p=0.002), Qwen3-4B (p=0.01), Qwen3-14B (p=0.008), and SmolLM3 (p=0.002) and not significant for Phi-4 (p=0.064).
>
> W3+Q2: In addition to the above significance tests, we run permutation tests (1000 resamples) to test the difference between our strongest baseline (NWKR) and each of the ACE methods on AUC EURO on APIGen for each model separately. We find that all differences are **statistically significant** at a significance level of 0.05. All p-values are less than 0.01. The smECE differences, where NWKR and HRE often outperform ACE, are not significant (p > 0.2). In the final version, we will include all significance tests for all tasks, configurations, and metrics across all models.
>
> W4: We will include requiring access to model internals in our limitations. Due to computational constraints, very large models are infeasible to run; we think that these experiments are unnecessary to support our conclusions, but based on your suggestion of evaluating a larger model and one that has an MoE architecture, we tested Qwen3-30B-A3B-Instruct-2507 on APIGen. Our results are slightly better on this model than the other 6 LLMs (ace pca20 has much lower smECE and much higher AUC EURO than baselines):
>
> | Estimator | smECE | AUC EURO (all) |
> | --- | --- | --- |
> | raw confidences | 0.263 | 0.510 |
> | NWKR | 0.042 | 0.797 |
> | ace pca20 | 0.027 | 0.868 |
> | ace mid | 0.057 | 0.870 |
> | ace late | 0.063 | 0.872 |
>
> Q3: For Qwen3-30B MoE on APIGen, generation took an average of 15.75 seconds per example on 2 L40S GPUs. The amortized time for inference on a trained random forest using ACE pca20 for an example is 0.00007 seconds, so the wall-clock overhead is nearly 0, even if our generation implementation can be sped up.
>
> Q4: SCITLDR is a very hard task. Mean accuracies across LLMs for different rouge-L thresholds (variances are < 0.003) are 0.1 - 93.8%, 0.2 - 60.0%, 0.3 - 22.1%, 0.4 - 5.6%, 0.5 - 1.0%, as a result, we evaluate with thresholds of 0.2, 0.3 (in the paper), and 0.4. ACE performs slightly better than nwkr at lower rouge-L settings:
>
> | Estimator | Threshold | smECE | AUC EURO (all) |
> | --- | --- | --- | --- |
> | NWKR | 0.2 | 0.183 | 0.743 |
> | pca20 | 0.2 | 0.184 | 0.752 |
> | ace mid | 0.2 | 0.180 | 0.755 |
> | NWKR | 0.4 | 0.016 | 0.887 |
> | pca20 | 0.4 | 0.026 | 0.886 |
> | ace mid | 0.4 | 0.018 | 0.889 |
>
> ACE methods also have greater probability spread, demonstrating better decision-making utility as with the 0.3 threshold in Figure 3.
>
> Other limitations: We disagree that the model range is narrow; we now test 7 models of varying sizes across model families and with an MoE model, finding similar trends across them. Our method is supervised with high sample complexity (strong performance with just 25 labeled examples, which are easy to collect) and our results are consistent across rouge-L thresholds.
>
> Thank you for helping us strengthen our work. We will add all of this to the final version, including running Qwen3-30B on the other tasks. If you have any other questions or concerns, we’d be happy to answer them. We would appreciate it if you could update your review taking our response into account.

---

> > ### Author Rebuttal · Reviewer_B8ZC · 2026-04-02
> >
> > Thank you for the detailed rebuttal. The additional experiments and statistical tests have addressed several of my concerns.
> >
> > The permutation tests (W3) now provide proper statistical backing. The Qwen3-30B MoE experiment (W4) is a convincing extension, and the noise interpolation ablation (W1) offers useful evidence that activations carry task-relevant signals—though a non-activation high-dimensional baseline would have been even more convincing.
> >
> >
> > SCITLDR gains remain small in absolute terms, but the threshold analysis and significance tests provide helpful context.
> >
> > Given these improvements, I raise my score.

---

> > > ### Author Response · Authors · 2026-04-05
> > >
> > > Thank you for your constructive feedback, engagement, and incorporation of our feedback into your review including raising your score.
> > >
> > > We agree that the gains on SCITLDR are small in absolute terms, but that the additional analysis provides context into how our methods do on this task.

---

### Decision · Program_Chairs · 2026-04-30

**Decision:**

Accept (regular)

**Comment:**

This paper proposes EURO, a risk-aware calibration metric, and ACE, an activation-based confidence estimation protocol for improving LLM calibration and decision-making utility .

Reviewers broadly agree that EURO addresses an important limitation of existing calibration metrics and that ACE demonstrates strong empirical performance, particularly in sample efficiency and decision utility. The work is technically sound with solid theoretical grounding and consistent improvements across multiple tasks and models. The rebuttal further strengthens the paper by adding statistical tests, additional baselines, and clarifications.

While some concerns remain regarding baseline strength, applicability to closed models, and limited gains on certain tasks, these do not undermine the core contributions.

Overall, I find this paper a meaningful contribution to LLM calibration and recommend acceptance.